# Diversity and determinants of recombination landscapes in flowering plants

**Thomas Brazier**[1], **Sylvain Glémin**[1,2,¤] *

**1** University of Rennes, CNRS, ECOBIO (Ecosystems, Biodiversity, Evolution), Rennes, France,
**2** Department of Ecology and Genetics, Evolutionary Biology Center and Science for Life Laboratory, Uppsala University, Uppsala, Sweden

¤ Current address: University of Rennes, CNRS, ECOBIO (Ecosystems, Biodiversity, Evolution)—UMR 6553, Rennes, France
* sylvain.glemin@univ-rennes1.fr

## Abstract

During meiosis, crossover rates are not randomly distributed along the chromosome and their location may have a strong impact on the functioning and evolution of the genome. To date, the broad diversity of recombination landscapes among plants has rarely been investigated and a formal comparative genomic approach is still needed to characterize and assess the determinants of recombination landscapes among species and chromosomes. We gathered genetic maps and genomes for 57 flowering plant species, corresponding to 665 chromosomes, for which we estimated large-scale recombination landscapes. We found that the number of crossover per chromosome spans a limited range (between one to five/six) whatever the genome size, and that there is no single relationship across species between genetic map length and chromosome size. Instead, we found a general relationship between the relative size of chromosomes and recombination rate, while the absolute length constrains the basal recombination rate for each species. At the chromosome level, we identified two main patterns (with a few exceptions) and we proposed a conceptual model explaining the broad-scale distribution of crossovers where both telomeres and centromeres play a role. These patterns correspond globally to the underlying gene distribution, which affects how efficiently genes are shuffled at meiosis. These results raised new questions not only on the evolution of recombination rates but also on their distribution along chromosomes.

## Author summary

Meiotic recombination is a universal feature of sexually reproducing species. During meiosis, crossovers play a fundamental role for the proper segregation of chromosomes during meiosis and reshuffles alleles among chromosomes. How much variation in recombination is expected within a genome and among different species remains a central question for understanding the evolution of recombination. We characterized and compared recombination landscapes in a large set of plant species with a wide range of genome size. We found that the number of crossovers varied little among species, from

**Data Availability Statement:** All data available to reproduce the results presented in the paper is available in a public data repository (https://doi.org/10.17605/OSF.IO/NUXD7).

**Funding:** SG received fundings from the Agence Nationale de la Recherche (ANR HotRec ANR-19-

CE12-0019-04). The funders had no role in study design, data collection and analysis, decision to publish, or preparation of the manuscript.

**Competing interests:** The authors have declared that no competing interests exist.

one mandatory to no more than five or six crossovers per chromosomes, whatever the genome size. However, we identified two main patterns of variation along chromosomes (with a few exceptions) that can be explained by a new conceptual model where chromosome length, chromosome structure and gene density play a role. The strong association between gene density and recombination was already known, but raised new questions not only about the evolution of recombination rates but also on their distribution along chromosomes.

## Introduction

Meiotic recombination is a universal feature of sexually reproducing species. New haplotypes are passed on to offspring by the reciprocal exchange of DNA between maternal and paternal chromosomes, known as crossovers (COs). However, recombination landscapes—the variation in recombination rates along the chromosome—are not homogeneous across the genome and vary among species [1–4]. Meiotic recombination involves chiasmata at pairing sites between homologous chromosomes to ensure the physical tension needed for the proper disjunction of homologs [1,3,5]. Recombination also plays an evolutionary role by breaking linkage disequilibrium between neighbouring sites and creating new genetic combinations transmitted to the next generation, making selection on individual genetic variant more efficient [6–8]. The number and location of crossovers along the chromosome are finely regulated through mechanisms of crossover assurance, interference and homeostasis [9,10]. In most species, crossover assurance is necessary to achieve proper segregation and to avoid deleterious consequences of nondisjunction, though it is not very clear whether at least one CO per chromosome or per arm is required. Additional COs are also usually regulated through interference, ensuring that they are not too numerous and not too close to each other [10,11]. In addition to regulation on a large scale [12,13], recombination is also finely tuned on a small scale. In plants studied so far, crossovers are concentrated in very short genomic regions (typically a few kb), i.e. recombination hotspots, which have been found in gene regulatory regions, and mostly in promoters [14–16].

In addition to their function in meiosis, variations in recombination rates affect genome structure, functioning and evolution through direct effects–such as mutagenic effects, bias-conversion, and ectopic exchanges–and indirect effects by modulating the efficacy of selection [17], and it has become a challenge to integrate recombination rate variation in population genomics in the age of 'genomic landscapes' [18,19]. The characterization of recombination landscapes also has practical interests as variation in meiotic genes could be used to experimentally manipulate CO patterns for purposes, such as redirecting recombination towards regions of interest for crop breeding [20].

In plants, recombination rates are believed to be higher in species with smaller genomes because the linkage map length is independent of genome size and the number of chromosomes explain more variation than genome size [4]. Several broad-scale determinants have recently been identified, such as chromosome length [21], distance to the telomere or centromere [22] and genomic and epigenetic features [16,23,24], notably the density of transposable elements (TEs), which is usually negatively correlated with recombination rates [25]. Plant genomes also contain large regions with suppressed recombination in various proportions (from a few Mb to hundreds of Mb, 1 to 75% of the genome). However, the diversity of recombination landscapes in plants still remain to be properly quantified.

Recently, a meta-analysis explored large-scale recombination landscapes among eukaryotes and paved the way for identifying general patterns [2]. They found that larger chromosomes have low crossover rates in their centre and suggested a simple telomere-led model with a universal bias of COs towards the periphery of the chromosome, positively correlated with chromosome length. They also proposed that chromosome length played the main role in crossover patterning while position of the centromere had almost no effect (except locally). Alternatively, it has also been proposed that both telomeres and centromeres shape recombination landscapes [26] and a universal pattern among plants has been questioned [13]. As only a limited number of species has been studied and as plant genomes are highly diverse in many ways [27,28], diversity in recombination landscapes may have been overlooked [29]. In addition, previous studies were meta-analyses combining heterogeneous datasets (ex: mix of inferred data from graphics, final processed data and only a few raw datasets in [2]) without a standard way to infer recombination maps, which prevented detailed comparisons among species.

To overcome these limitations we gathered the largest recombination landscape dataset in flowering plants, to the best of our knowledge. We started from raw data by combining genetic mapping from pedigree data and chromosome scale genome assemblies, from which we estimated recombination maps–more precisely the sex-averaged rate of COs along chromosomes–using the same standardised method in all species, in order to ask the following questions. What is the range of COs per chromosome in plants? Is the distribution of COs shaped by genome structure (i.e. chromosome size, telomeres, centromeres) and if so is there a universal pattern? Since recombination is negatively associated with TEs and recombination hotspots have been found in gene regulatory, are recombination landscapes always associated with gene density? What are the consequences of recombination heterogeneity on the extent of genetic shuffling? Overall, we found that recombination landscapes in plants are more diverse and more complex than previously thought. We identified two main patterns that are correlated with, and which may emerge from, the gene density distribution. We show that the positive association between gene density and recombination rates globally improves the genetic shuffling of coding regions, which raises new questions about the evolution of recombination.

## Results

### Dataset and recombination maps

We retrieved publicly available data for sex-averaged linkage maps and genome assemblies to obtain genetic and physical distances. We selected linkage maps for which the markers had genomic positions on a chromosome-level genome assembly (except for *Capsella rubella*, which had a high-quality scaffold-level assembly of pseudo-chromosomes). We remapped markers on the reference genome for 14 species for which genomic positions were not known or were mapped to an older assembly. After making a selection based on the number of markers, marker density, and genome coverage, and after filtering out the outlying markers (see methods), we produced 665 chromosome-scale Marey maps (plots of genetic vs physical distances, expressed as cM vs Mb) for 57 species (2–26 chromosomes per species, S1 and S2 Tables and S1 and S2 Figs). The number of markers per chromosome ranged from 31 to 49,483, with a mean of 956 markers. Correcting the linkage map lengths (Hall & Willis's method) did not change the total linkage map lengths (mean difference = 1.19 cM, max difference = 5.62 cM), giving confidence in the coverage of the linkage map [30]. We verified that neither the number of markers, marker density nor the number of progenies had a significant effect on the analyses (S3 Fig). We also retrieved gene annotations for 41 genomes. The

angiosperm phylogeny was well represented in our sampling (S4 Fig), with a basal angiosperm species (*Nelumbo nucifera*), 15 monocot species and 41 eudicots. From the literature, we also obtained data on the centromeric index for 37 species, defined as the ratio of the short arm length divided by the total chromosome length (S3 Table).

From the Marey maps, we estimated local recombination rates along the chromosomes in non-overlapping 100 kb windows, and their 95% confidence intervals (1,000 bootstraps). Estimates at a scale of 1 Mb yielded very similar results (the Spearman rank correlation coefficient correlation between the two estimates was Rho = 0.99, p < 0.001, S4 Table) therefore only 100 kb landscapes were analysed in the subsequent analyses.

## Smaller chromosomes have higher recombination rates than larger ones

Mean genome-wide recombination rates spanned two orders of magnitude, ranging from 0.2 cM/Mb in bread wheat (*Triticum aestivum*) to 16.9 cM/Mb in squash (*Cucurbita maxima*). In agreement with previous studies [2,4], we found a significant negative correlation between chromosome size (Mb) and the mean chromosomal recombination rate (Spearman rank correlation coefficient Rho = -0.84, p < 0.001; log-log Linear Model, adjusted $R^2$ = 0.83, p < 0.001). Most species had, on average, between one and four COs per chromosome although the genome sizes span almost two orders of magnitude. Less than 2% of chromosomes had less than one CO on average (n = 11). 234 chromosomes had between one and two COs on average, suggesting that a single CO per chromosome is sufficient, though 419 chromosomes had more than two COs. However, as genetic maps are based on the average of several meiosis, they do not give access to the distribution of COs per meiosis. Thus, it is worth noting that chromosome genetic maps higher than 50 cM do not imply that all chromosome always exhibit at least one CO.

Using a Linear Mixed Model we found a significant species random effect that explained 82% of the variance ($\log_{10}$(recombination rate) ~ $\log_{10}$(chromosome size) + (1 | species), marginal $R^2$ = 0.17, conditional $R^2$ = 0.96, p < 0.001). Adding phylogenetic covariance did not improve the mixed model, so we did not retain a phylogenetic effect (S5 Table). Interestingly, the (log-log) relationship between the recombination rate and chromosome size was not the same within and between species, suggesting that absolute chromosome size does not have the same effect in all species (Fig 1B). Similarly, the relationship between linkage map length (cM) and chromosome size (Mb) was highly species-specific (linkage map length ~ $\log_{10}$(chromosome size) + (1 | species), marginal $R^2$ = 0.49, conditional $R^2$ = 0.99, p < 0.001) (Fig 2A), with species slopes decreasing with the mean chromosome size in a log-log relationship. It indicates that species slopes are roughly proportional to the inverse of the mean chromosome size (S5 Fig). Consequently, the excess of COs on a chromosome (i.e. the linkage map length minus 50 cM) was correlated with its relative size (i.e. chromosome size divided by the mean chromosome size of the species; Fig 2B) not the absolute size. Moreover, in contrast to the relationship between recombination rate and absolute size, we did not observe any difference between the linear model and the fixed regression of the mixed linear model, suggesting that this relationship is similar across species (Fig 2B). In other words, any two chromosomes with the same ratio of sizes will have the same ratio of excess of recombination rate, whatever the species and the genome size.

## Diversity of CO patterns among flowering plants

Recombination landscapes along chromosomes appeared to be qualitatively very similar within species but strongly varied between species (Figs 3 and S2). In the text below, to avoid confusion with the molecular composition and specific position defining telomeric and

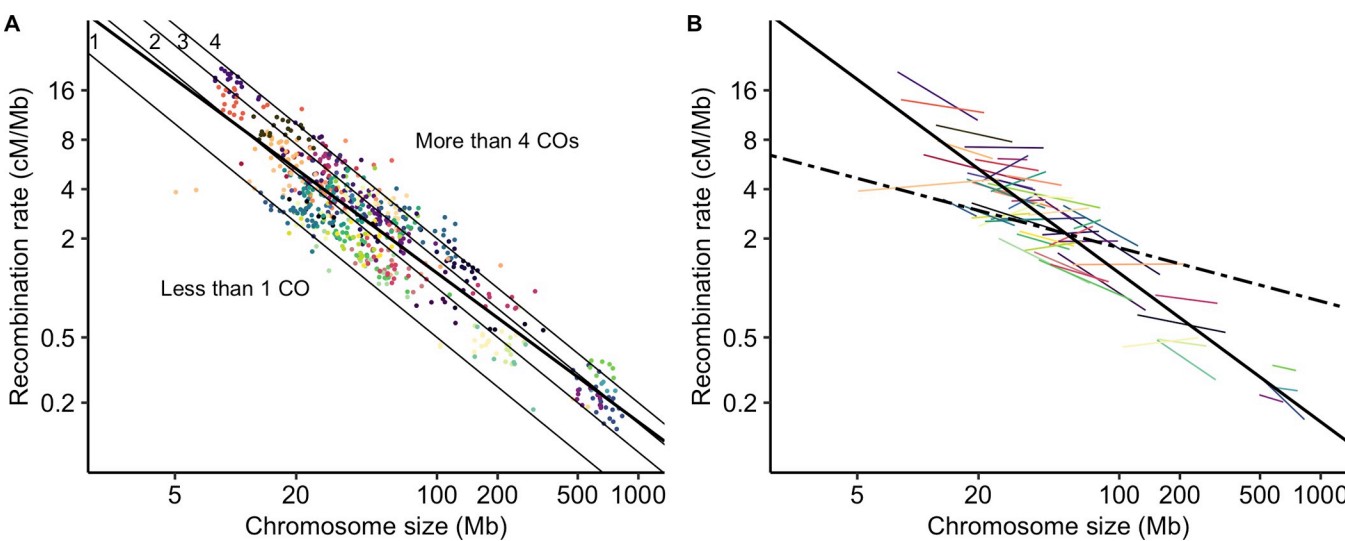

**Fig 1. Mean recombination rates per chromosome (cM/Mb, log scale) are negatively correlated with chromosome physical size (Mb, log scale).** Each point represents a chromosome (n = 665). Species are presented in different colours (57 species). (A) The bold solid line represents the linear regression line fitted to the data. The thin lines correspond to the expectation of one, two, three or four COs per chromosome. (B) Correlations between recombination rates and chromosome size within each species with at least 5 chromosomes (coloured lines, 55 species) and the overall between-species correlation controlled for a species effect (black dashed line, n = 57 species). Solid bold line as in (A).

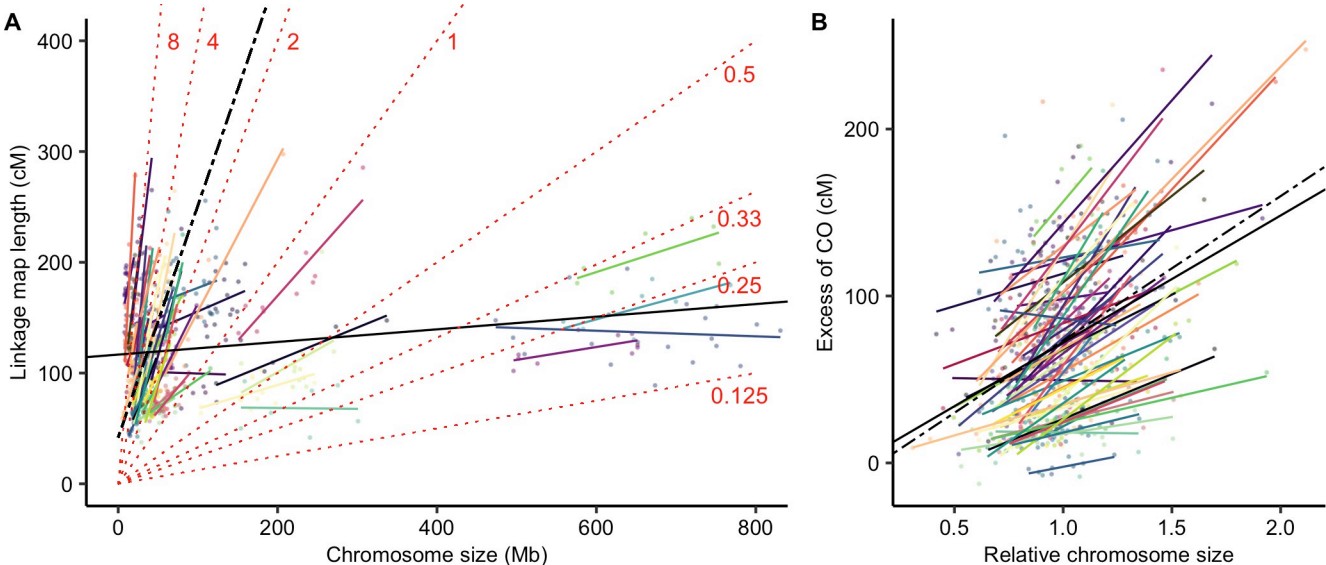

**Fig 2. Linkage map length (cM) is positively correlated with genomic chromosome size (Mb).** (A) Correlation between chromosome genomic size (Mb) and linkage map length (cM). Each point represents a chromosome (n = 665). Species are presented in different colours (57 species). The black solid line represents the simple linear regression (linkage map length ~ $\log_{10}$(chromosome size), adjusted $R^2$ = 0.036, p < 0.001) and the black dashed line the fixed effect of the mixed model (linkage map length ~ $\log_{10}$(chromosome size) + (1 | species), marginal $R^2$ = 0.49, conditional $R^2$ = 0.99, p < 0.001). Species random slopes are shown in colours. Isolines of recombination rates are plotted for different values (indicated cM/Mb) as dotted red lines to represent regions with equal recombination. (B) The excess of COs (linkage map length minus 50 cM for the obligate CO) is positively correlated with the relative chromosome size (size / average size of the species). The black solid line is the linear regression across species (excess of CO ~ relative chromosome size, adjusted $R^2$ = 0.13, p < 0.001) and the black dashed line the fixed effect of the mixed model (excess of CO ~ relative chromosome size + (1 | species), marginal $R^2$ = 0.14, conditional $R^2$ = 0.86, p < 0.001). Coloured solid lines represent individual regression lines for species with at least 5 chromosomes (55 species).

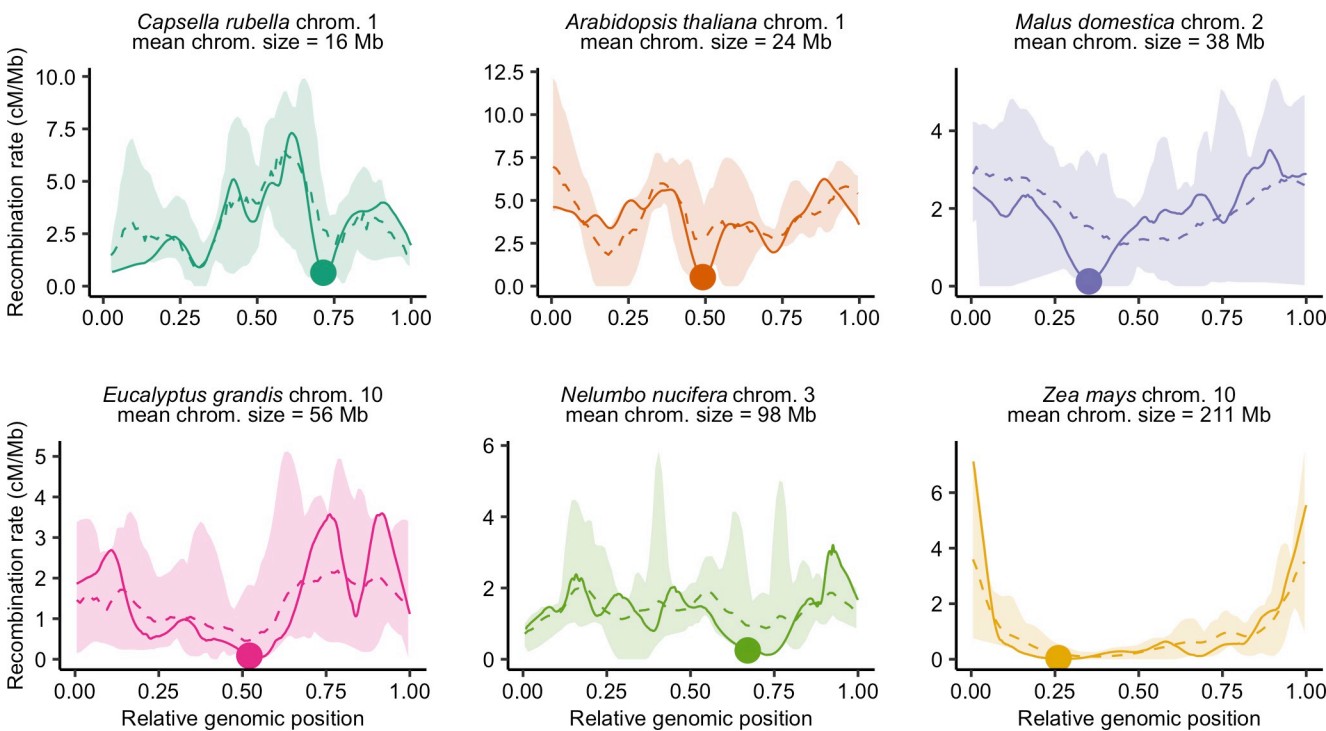

**Fig 3. Diversity of recombination landscapes exemplified by six different species.** Recombination landscapes are similar within species (the dashed line is the average landscape for pooled chromosomes, all recombination landscapes of the species are contained within the colour ribbon). Genomic distances (Mb) were scaled between 0 and 1 to compare chromosomes with different sizes. Estimates of the recombination rates were obtained by 1,000 bootstraps over loci in windows of 100 kb with loess regression and automatic span calibration. One chromosome per species is represented in a solid line, with the genomic position of the centromere demarcated by a dot. The six species are ordered by ascending mean chromosome size (Mb).

centromeric regions *stricto sensu*, we have used instead the terms distal regions for the extremities of the chromosomes and proximal regions for the central part of the chromosomes around the centromere. Note that in the species we surveyed none had acrocentric chromosome. Representing relative recombination rates in ten bins of equal physical length (see Materials and Methods for details), some landscapes appeared rather homogeneous along chromosomes whereas others were extremely structured with recombination concentrated in the short distal parts of the genome, and wide variations between these two extremes (Fig 4). The Gini index is a measure of heterogeneity bounded between 0 (perfect homogeneity) and 1 (maximal heterogeneity). The range of the Gini index estimated on recombination landscapes was between 0.1 and 0.9 (S1 Table). The bias towards the periphery was not ubiquitous across species (Fig 4), whereas Haenel et al. [2] suggested that the distal bias could be universal for chromosomes larger than 30 Mb. Only a subset of species, especially those with very large chromosomes (> 100 Mb), exhibited a clear bias (Fig 4). Despite large chromosome sizes (mean chromosome sizes = 98 Mb and 222 Mb, respectively), *Nelumbo nucifera* and *Camellia sinensis* are noticeable exceptions to this pattern, with the highest recombination rates found in the middle of the chromosomes (*Nelumbo nucifera* illustrated in Fig 3E, other species in S2 Fig). For small to medium-sized chromosomes, the pattern is less clear. Most species did not show any clear structure along the chromosome but a few of them (e.g. *Capsella rubella*, *Dioscorea alata*, *Mangifera indica*, *Manihot esculenta*) showed a drop in recombination rates in the distal regions and high recombination rates in the proximal regions (*Capsella rubella* illustrated in Fig 3A).

Following Haenel et al. [2], we calculated the periphery-bias ratio as the recombination rate in 10% at each extremity of each chromosome divided by the mean recombination rate. A ratio higher than 1 indicates a higher recombination rate in the tips than the whole chromosome. By pooling chromosomes within species, the periphery-bias ratio ranged from 0.31 to 5.76. Most species exhibited a ratio higher than one but the ratio was lower than one for nine species and just above one (<1.2) for six species (see S5 Table). We detected a significant positive effect of chromosome length on the periphery-bias ratio across species (Linear Model, adjusted $R^2$ = 0.44, p < 0.001; Fig 5A) with some exceptions (see *Capsella rubella* and *Nelumbo nucifera* on Fig 3). Analysing chromosomes independently across all species the mean periphery-bias ratio is significantly higher than 1 (95% bootstrapped confidence interval of the mean = [2.06;2.32]) and skewed towards values higher than 1 but the correlation with chromosome length within each species was not clear (Fig 5B, 5C and S6 Table).

## Joint effect of telomeres and centromeres on crossover distribution along chromosomes

Globally, recombination rates were negatively correlated with the distance to the nearest telomere (S6 Fig and S7 and S8 Tables). However, two qualitative patterns emerged (Figs 6 and S7, S8 Table). In 34 species, recombination decreased from the telomere and reached a plateau at approximately 20% of the whole chromosome length (the distal pattern, Fig 6A), in agreement with the model suggested by Haenel et al. [2]. Sixteen species exhibited a sharp decrease in the most distal regions and a peak of recombination in the sub-distal regions (relative genomic distance between 0.1–0.2) followed by a slow decrease towards the centre of the chromosome (the sub-distal pattern, Fig 6B). There were a few exceptions to these two patterns (six species), e.g. *Capsella rubella* consistently showed higher recombination rates in the middle of the chromosome (Fig 3A). Interestingly, species classified as having a distal pattern had significantly larger chromosomes than species classified as having a sub-distal pattern (Wilcox rank sum test, p < 0.001, Fig 6C). Furthermore, the correlation between recombination and the distance to the nearest telomere was significantly higher for species with larger chromosomes (Spearman rank correlation coefficient Rho = -0.51, p < 0.001; S6 Fig).

When the centromere position was known, we confirmed that the centromeres had an almost universal local suppressor effect (Figs 3 and 4). In small and medium-sized chromosomes, the recombination was often suppressed only in short restricted centromeric regions (several Mb, 1–5% of the map) displaying drastic drops in the recombination rates. In larger chromosomes, the suppression of recombination extends to large regions upstream and downstream of the physical centre of the chromosome (approximately 80–90% of the chromosome; Fig 4). Ninety percent of chromosomes (388 chromosomes) had significantly less recombination than the chromosome average at the centromeric index (n = 425, resampling test, 1,000 bootstraps, 95% confidence interval). In 81 chromosomes (19%) estimated recombination was null in the centromere (although it can be non-zero but lower than the detection threshold given the number of individuals used for genetic maps). However, the transposition of centromere position from cytological data to genomic data may be imprecise or wrongly oriented for some chromosomes. After orienting chromosomes to map the centromeric index, 16% of chromosomes (70 over 425) had a recombination rate slightly higher in the inferred centromere position than on the opposite side, thus a centromere potentially mapped on the wrong side.

To understand the patterns observed further, we compared three models (Fig 7). Under the strict distal model proposed by Haenel et al. [2] (M1), the centromere plays no role beyond its local suppressor effect, which predicts an equal distribution of crossovers on both sides of the

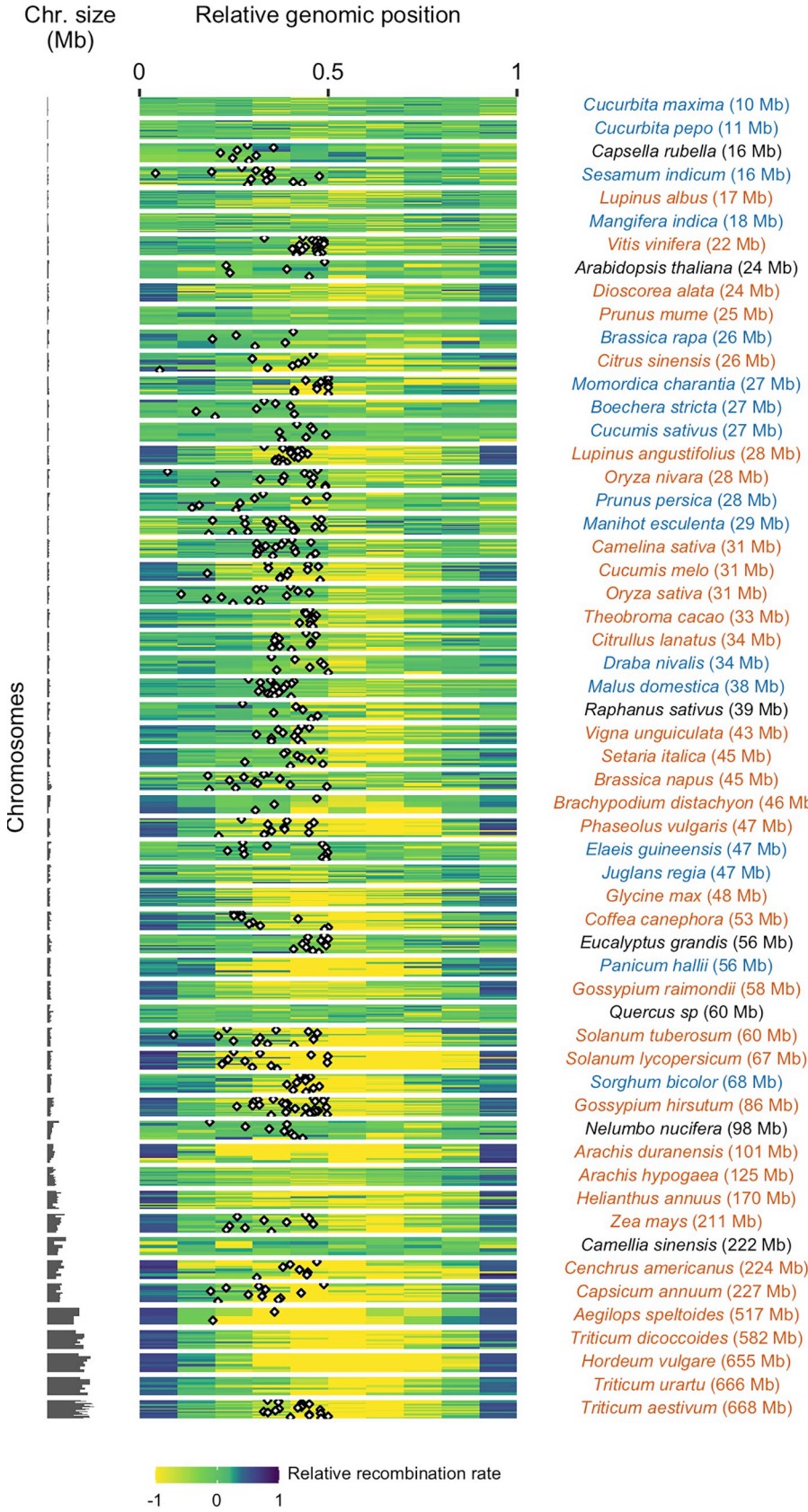

**Fig 4. Patterns of recombination within chromosomes (n = 665).** Relative recombination rates along the chromosome were estimated in ten bins of equal ratio of the observed genetic length divided by the expected genetic length (one tenth of total size) of the bin (log-transformed). Values below (above) zero are recombination rates that are lower (higher) than expected under a random distribution. The 57 species are ordered by ascending genome size. Each horizontal bar plot represents one chromosome. When available, the centromere position is mapped as a black and white diamond.

centre of chromosomes, independently of centromere position: $\frac{d(1/2)}{d(1)} = 0.5$, where $d(1/2)$ is the genetic distance (cM) to the physical middle of the chromosome and $d(1)$ is the total genetic distance (cM). We also tested two alternative models with a centromere effect. We assumed that the position of the centromere, $d(c)$, affects the distribution of crossovers along the chromosome. Models M2 'telomere + centromere + one CO per arm' and M3 'telomere + centromere + one CO per chromosome'; both assume that the relative genetic distance of a chromosome arm is proportional to its relative genomic size. However, they differ in the number and distribution of mandatory COs. In M2, at least one CO in each chromosome arm (50 cM) is mandatory whereas only one CO is mandatory for the entire chromosome in M3, even if it has two arms (which always the case in our dataset). For species whose centromere position was known (37 species, 425 chromosomes) we regressed the observed values against the theoretical predictions of the three models and compared them using goodness-of-fit criteria (adjusted $R^2$, AIC, BIC). M1 was not supported by any species and M2 was generally rejected

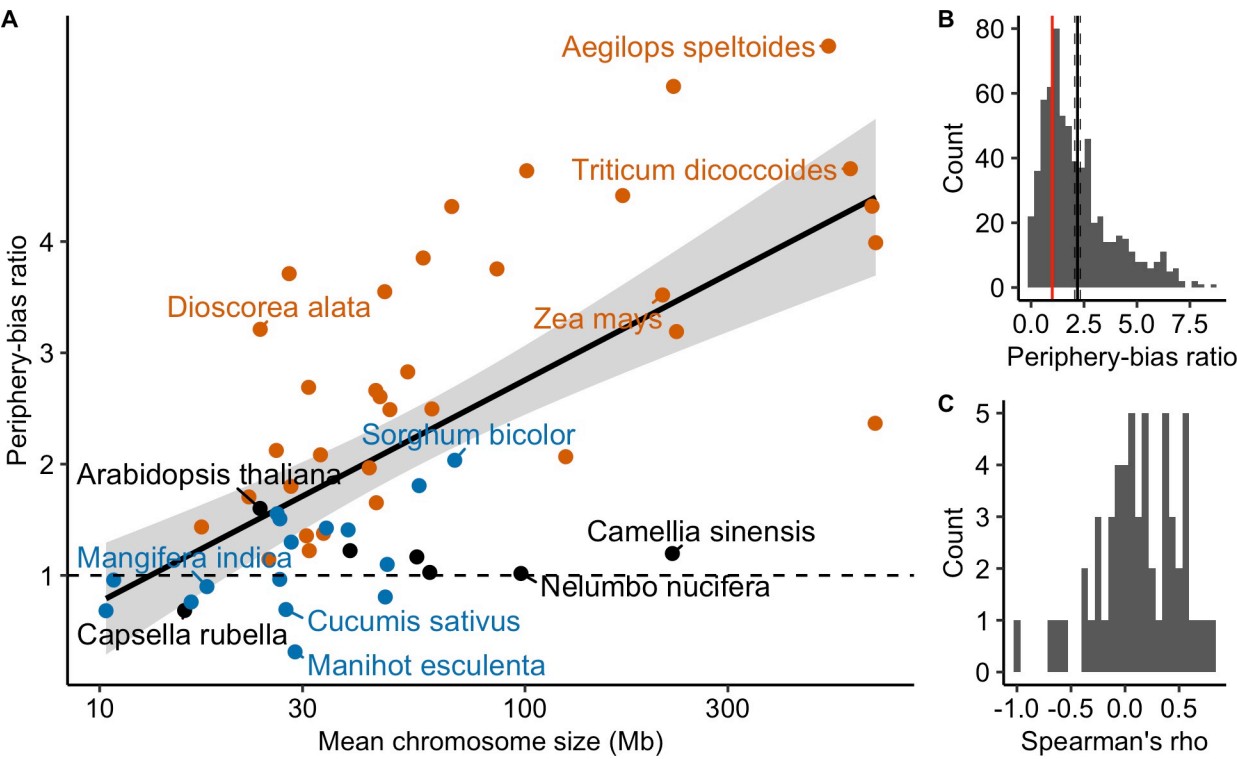

**Fig 5. The periphery-bias ratio is positively correlated with chromosome genomic size.** (A) Linear regression between the species mean periphery-bias ratio and the mean chromosome size (log scale) across species (n = 57 species; adjusted $R^2$ = 0.44, p < 0.001). Points are coloured according to the classification of the CO patterns described below (orange = distal, blue = sub-distal, black = unclassified). (B) Distribution of periphery-bias ratios (n = 665 chromosomes). The mean periphery-bias ratio and its 95% confidence interval (black solid and dashed lines) were estimated by 1,000 bootstrap replicates. The red vertical line corresponds to a ratio of one. (C) Distribution of Spearman's correlation coefficients between the periphery-bias ratio and chromosome genomic size (Mb) within species (n = 57 species).

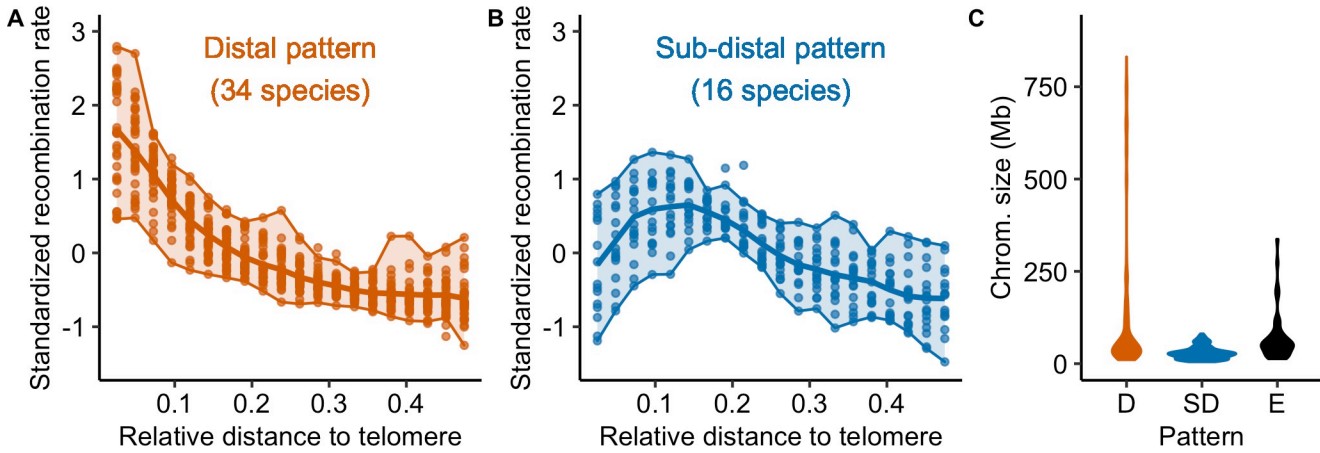

**Fig 6. Distribution of crossover: main patterns.** (A and B) Standardized recombination rates for species (chromosomes pooled per species, n = 57 species) are expressed as a function of the relative genomic distance from the telomere in 20 bins representing the two main patterns (orange = distal, blue = sub-distal). The seven unclassified species are shown in supplementary (S7 Fig). Chromosomes were split in half and 0.5 corresponds to the centre of the chromosome. In each plot, the solid line represents the mean recombination rate estimated in a bin (20 bins) and each dot per bin represents the average of a species. Upper and lower boundaries of the ribbon represent the maximum and minimum values. (C) Distribution of chromosome genomic sizes (Mb) for each pattern (D: distal, SD: sub-distal, E: exceptions).

since 22% of chromosomes had genetic maps lower than 50 cM in at least one arm, even though it was supported in a handful of species (Table 1). M3 was the best supported model (30 out of 37 species), with good predictive power (Spearman rank correlation between predicted and observed values: Rho = 0.72, p < 0.001; Tables 1, S9 and S10). Given that chromosome arm genetic maps shorter than 50 cM are incompatible with one mandatory CO per arm in model M2, we also compared the three models on a subset of chromosomes with at least 50 cM on each chromosome arm (n = 36 species, 333 chromosomes) which confirmed that model M3 was the best model. Similarly, we reran the model using only chromosomes whose centromere positions were known with certainty (n = 37 species, 355 chromosomes) and found the same results.

## Recombination rates are positively correlated with gene density

It has been shown in a few species that COs preferentially occur in gene promoters. The scale of 100 kb used here is too large to test whether this pattern is shared among angiosperms.

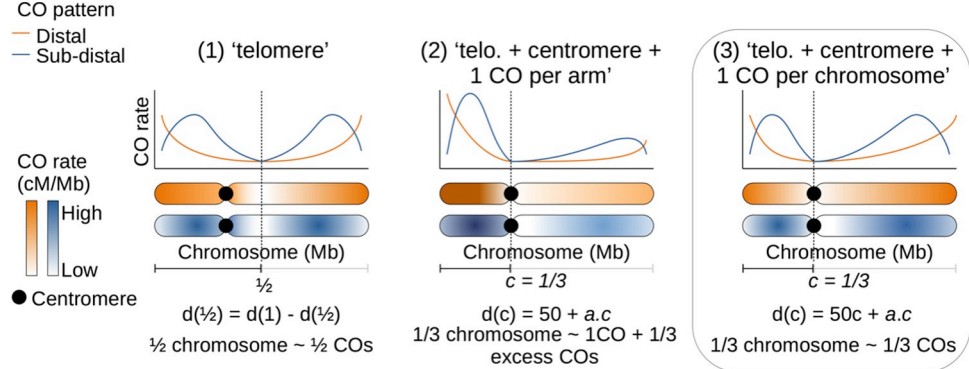

**Fig 7. Possible models of crossover patterns.** Schematic representation of the three competing models for the two main patterns, with an example of a centromere position at 1/3 of the chromosome. Model 3 is the best model (box).

**Table 1. Model selection for the telomere/centromere effect (n = 37 species with a centromere position, 425 chromosomes).** Three competing models were compared based on the adjusted $R^2$, p-value and AIC-BIC criteria among chromosomes (the best supported model is in bold characters). The number of species supporting each model was calculated based on the adjusted $R^2$ within species, for all species with at least five chromosomes. (1) 'telomere' model. (2) 'telomere + centromere + one CO per arm' model. (3) 'telomere + centromere + one CO per chromosome' model. $d(c)$ is the genetic distance to the centromere. $d(1)$ is the total genetic distance. A second model selection was done on a subset of chromosomes with at least 50 cM on each chromosome arm (n = 36 species, 333 chromosomes).

| # | Model | Expected | Adjusted $R^2$ | p | AIC | BIC | Species |
|---|---|---|---|---|---|---|---|
| | | Full dataset (37 species, 425 chromosomes) | | | | | |
| 1 | Telomere | d(1/2) / d(1) = 0.5 | 0.22 | < 0.001 | -477.8 | -465.7 | 0 |
| 2 | Tel. + Cent. + CO per arm | (d(c)– 50) / (d(1)– 100) = c | - | 0.72 | 3098.2 | 3110.4 | 7 |
| 3 | Tel. + Cent. + CO per chr. | d(c) / d(1) = c | 0.51 | < 0.001 | -476.6 | -464.5 | 30 |
| | | Subset (36 species, 333 chromosomes) | | | | | |
| 1 | Telomere | d(1/2) / d(1) = 0.5 | 0.18 | < 0.001 | -407.5 | -396.1 | 0 |
| 2 | Tel. + Cent. + CO per arm | (d(c)– 50) / (d(1)– 100) = c | -0.001 | 0.42 | 1939.1 | 1950.5 | 10 |
| 3 | Tel. + Cent. + CO per chr. | d(c) / d(1) = c | 0.50 | < 0.001 | -396 | -384.6 | 26 |

Instead, like in Haenel et al. [2], we assessed whether recombination increased with gene density. This pattern is also predicted if there is a negative association between TEs and recombination. Forty-one genomes were annotated with gene positions. Across chromosomes, the distribution of chromosomal correlations between gene count and recombination rate was clearly skewed towards positive values, independently of the previously described CO patterns (mean Spearman's rank correlation = 0.46 [0.43; 0.49]; Fig 8A). Ninety-one percent of 483 chromosomes (41 species) showed a significant correlation between the number of genes and recombination rate at a 100 kb scale. The strength of the relationship greatly varied across species and did not correlate with chromosome length or the genome-wide recombination rate

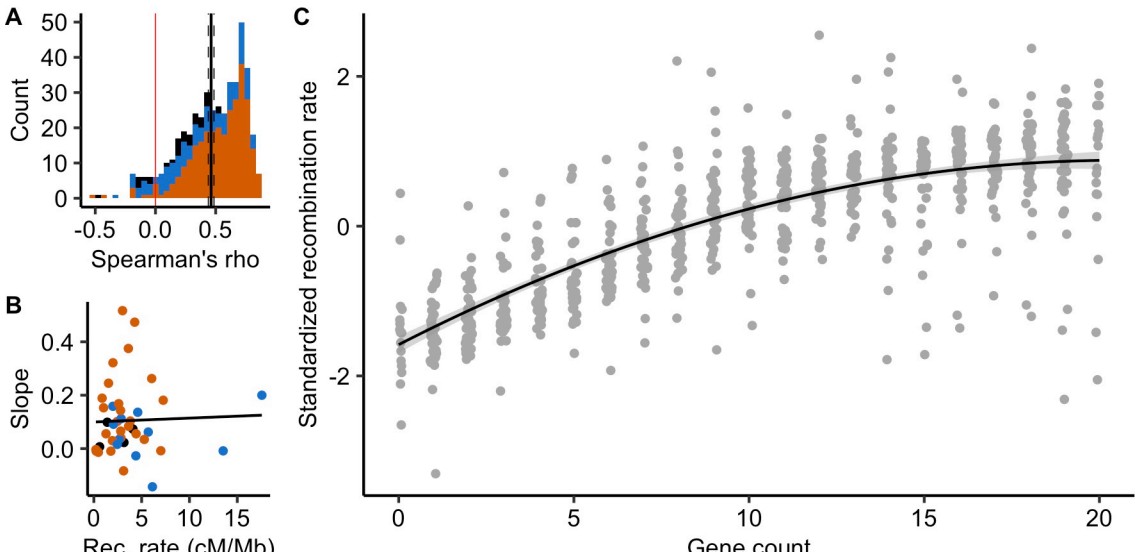

**Fig 8. Recombination rates are positively correlated with gene density (n = 483 chromosomes, 41 species).** (A) Distribution of chromosome Spearman's rank correlations between the number of genes and the recombination rate in 100 kb windows. The black vertical line is the mean correlation with a 95% confidence interval (dashed lines) estimated by 1,000 bootstrap replicates. Colours correspond to CO patterns (orange = distal, blue = sub-distal, black = exception). (B) Slopes of the species linear regression between gene count and recombination rates are independent of the species averaged recombination rate (Linear Model, adjusted $R^2$ = -0.02, p = 0.83). (C) Standardized recombination rates for each number of genes in a 100 kb window (centred-reduced, chromosomes pooled per species) estimated by 1,000 bootstraps and standardized within species. The gene count was estimated by counting the number of gene starting positions within each 100 kb window. The black line with a grey ribbon is the quadratic regression estimated by linear regression with a 95% parametric confidence interval (Linear Model, adjusted $R^2$ = 0.62, p < 0.001).

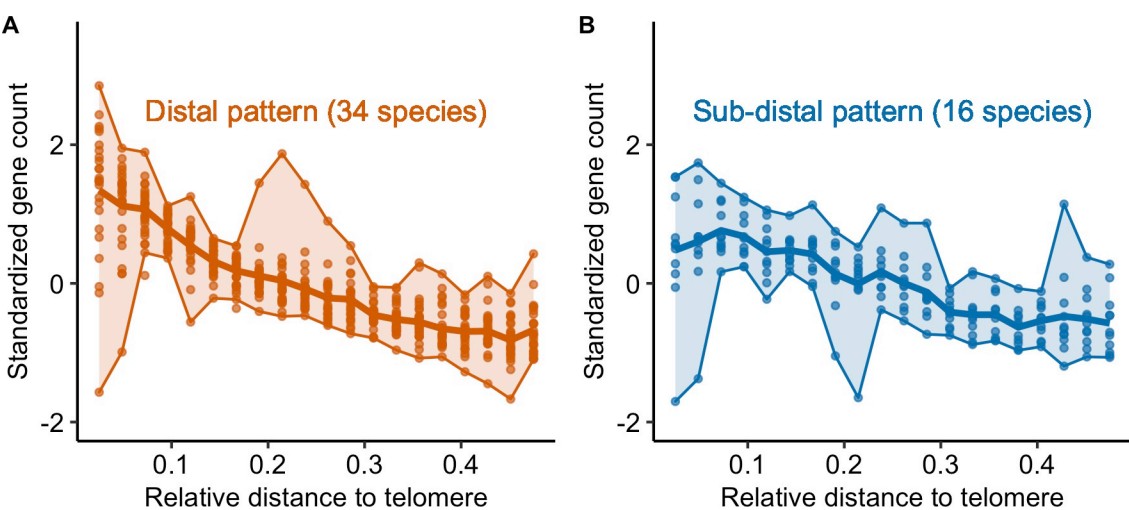

**Fig 9. Gene counts patterns along the chromosome are correlated with CO patterns (n = 41 species).** Standardized gene count (centred-reduced) as a function of the relative distance from the tip to the middle of the chromosome (genomic distances distributed in 20 bins). We used the same groups as identified for the CO pattern in Fig 6; (A) distal pattern vs (B) sub-distal pattern. Same legend as Fig 6.

(Fig 8B). Overall, standardized recombination rates (subtracting the mean and dividing by the standard deviation to allow comparison among species) consistently increased with the number of genes in most species (linear quadratic regression, adjusted $R^2 = 0.62$, $p < 0.001$; Fig 8C).

As for recombination patterns, we classified patterns of gene density along chromosomes in three categories: distal, sub-distal and exceptions (S8 Fig). Most species (30 out of 41) were classified in the same gene density and recombination pattern (S11 Table). Moreover, we observed the same qualitative pattern for gene density and recombination for species with either major recombination pattern (Fig 9).

## Quantification of genetic shuffling

We confirmed that crossovers are unevenly distributed in genomes, which should affect how genetic variation is recombined between parental homologous chromosomes. Recently, Veller et al. [31] proposed a measure to quantify the amount of genetic shuffling within and among chromosomes. To quantify how much it depends on the distribution of COs, we estimated its intrachromosomal component, $\bar{r}_{intra}$, as described in equation 10 in Veller et al. [31]. The $\bar{r}_{intra}$ gives, for a chromosome, a measure of the probability for a random pair of loci to be recombined by a crossover. As expected, this was positively and significantly correlated with linkage map length ($\bar{r}_{intra}$ ~ linkage map length + (1 | species), marginal $R^2 = 0.43$, conditional $R^2 = 0.88$, $p < 0.001$, S9 Fig). A pattern in which COs are physically clustered in distal chromosome regions is thought to generate less recombination than one with COs evenly distributed across the chromosome [31]. At a chromosomal level, consistently (across species) the periphery-bias ratio has a low but significant effect on genetic shuffling measure, consistent among species ($\bar{r}_{intra}$ ~ periphery-bias ratio + (1 | species), marginal $R^2 = 0.05$, conditional $R^2 = 0.68$, $p < 0.001$, S10 Fig). The more COs are clustered in the tips of the chromosome, the lower the chromosomal genetic shuffling. These results verify the analytical predictions of Veller et al. [31], although the strength of the effect remains weak.

However, the distributions of COs and genes are both non-random and often correlated (Figs 8 and S11). Genomic distances measured in base pairs may not be the most appropriate measure of genetic shuffling among functional genomic components. Thus, we also measured

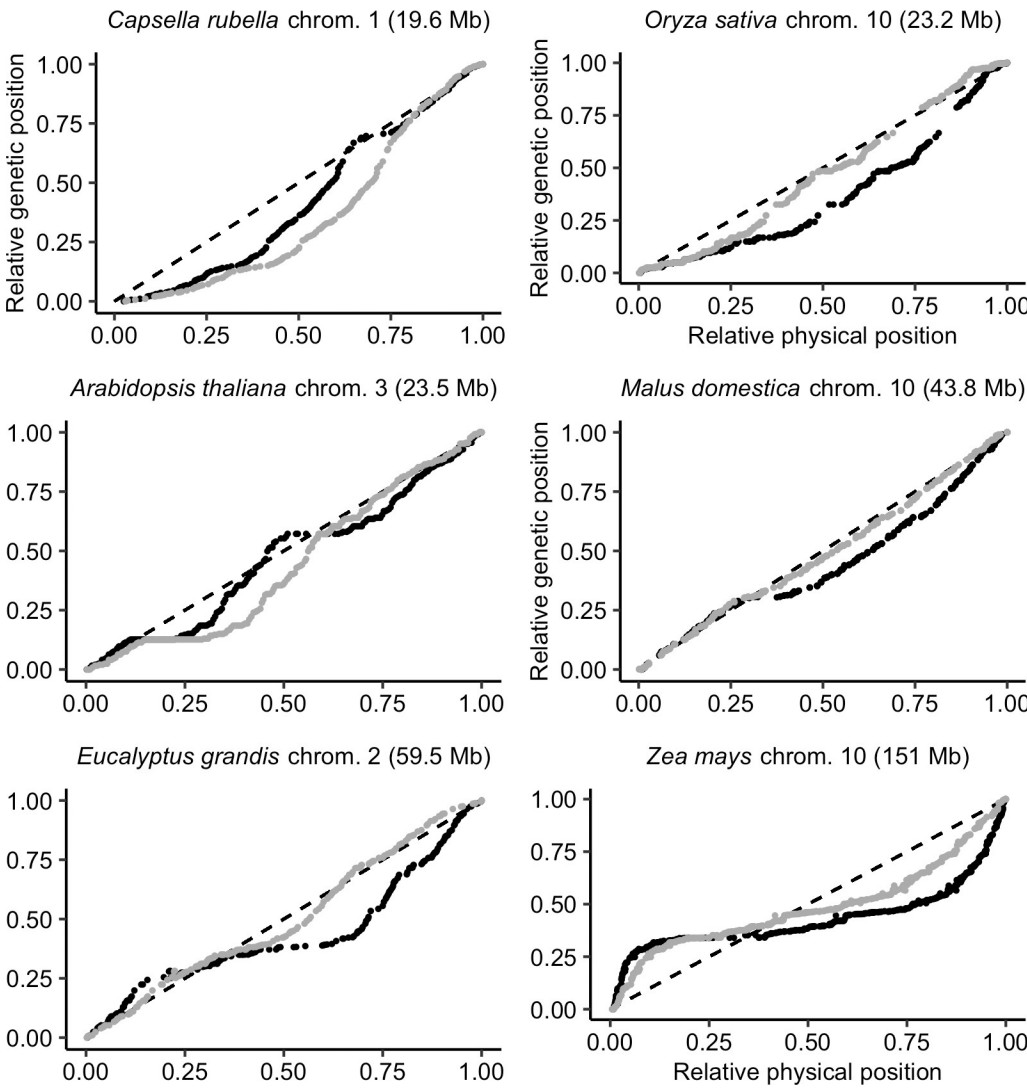

**Fig 10. Marey maps of six chromosomes with the relative physical distance expressed in genomic distances (black dots, position in the genome in Mb) or in gene distances (grey dots, position measured as the cumulative number of genes along the chromosome).** Marey maps are ordered by ascending chromosome size (Mb). The diagonal dashed line represents a theoretical random distribution of COs along the chromosome.

genomic distances in gene distances (i.e. the cumulative number of genes along the chromosome) instead of base pairs. Marey maps most often appeared more homogeneous when scaled on gene distances instead of base pair distances, with 70% (316 over 450) of Marey maps showing a smaller departure from a random distribution (Figs 10 and S12, S11 Table). This is not an automatic effect of changing scale as we compared the two maps on relative scales. Globally, a subset of 30 species has more homogeneous Marey maps with gene distances whereas 11 others are quantitatively more heterogeneous (notably *Capsella rubella* and *Arabidopsis thaliana*), although this could be due to low quality annotations making it difficult to precisely estimate the gene distances for some of them (e.g. *Sesamum indicum*). In most cases, genetic shuffling measures were slightly higher when gene distances were used instead of base pairs (Fig 11; mean = 0.22 for base pairs; mean = 0.26 for gene distances; Wilcoxon rank sum test with continuity correction, p < 0.001), implying more recombination among coding regions than

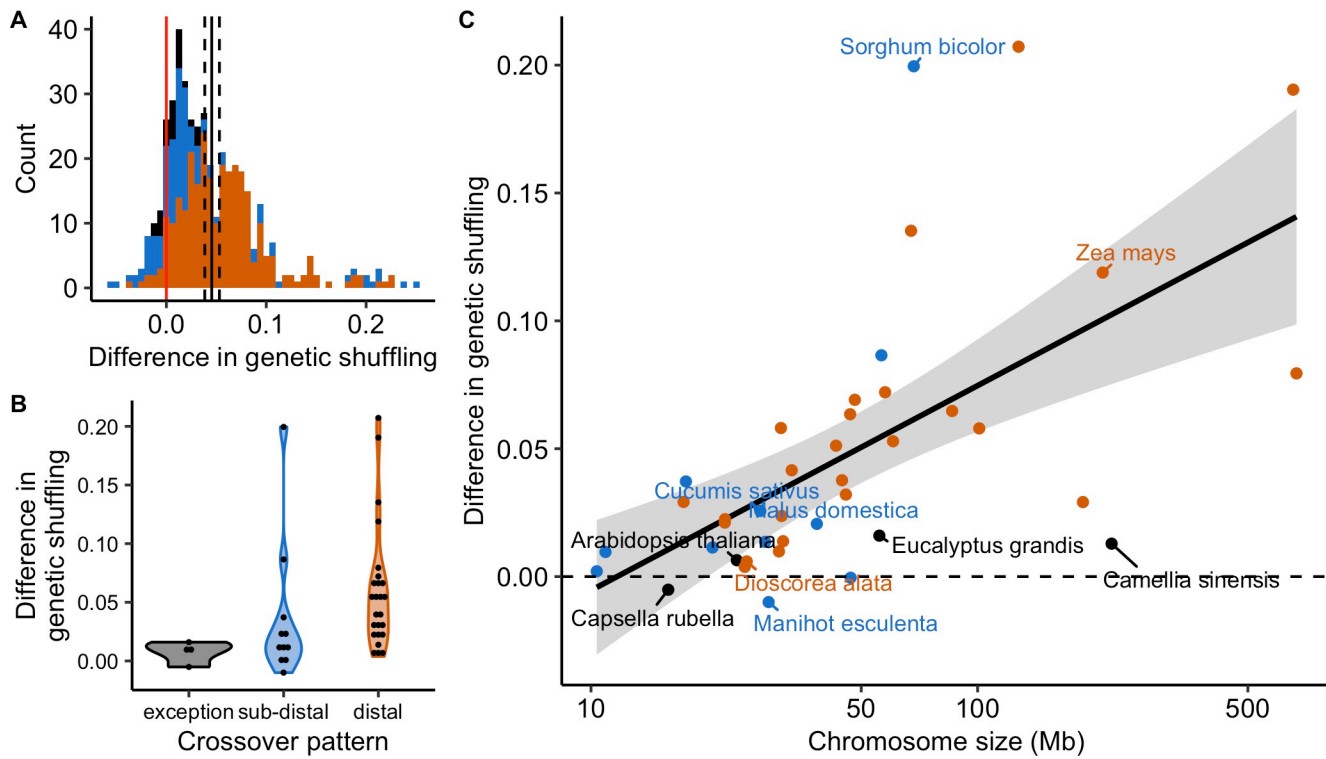

**Fig 11. Differences in genetic shuffling between estimates based on genomic distances (Mb) and gene distances (cumulative number of genes).** The difference is the genetic shuffling in gene distances minus the genetic shuffling in genomic distances. Colours correspond to CO patterns (orange = distal, blue = sub-distal, black = exception). (A) Distribution of the chromosome differences in the genetic shuffling (n = 444 chromosomes). (B) Distributions of the species difference in the genetic shuffling (n = 41 species, chromosomes pooled). (C) Species differences in the genetic shuffling are positively correlated with the averaged chromosome size (Linear Model, adjusted $R^2$ = 0.20, p = 0.002, n = 41, 95% parametric confidence interval).

among regions randomly sampled in the genome. Interestingly, the increase in genetic shuffling calculated in gene distances compared to genomic distance was more pronounced for longer chromosomes—which are often the most heterogeneous ones, characterized by a distal pattern—whereas we saw little effect on smaller chromosomes characterized by a sub-distal pattern (difference in $\bar{r}_{intra}$ ~ $\log_{10}$(chromosome size) + (1 | species), marginal $R^2$ = 0.21, conditional $R^2$ = 0.87, p < 0.001, Fig 11).

## Discussion

Based on a large and curated dataset, we provided a broad survey of recombination landscapes among flowering plants. In addition to confirming that both the chromosome-wide recombination rate and the heterogeneity of recombination landscapes vary according to chromosome length, we identified two distinct CO patterns and we proposed a new model that extended the strict telomere model recently proposed by Haenel et al. [2]. Moreover, the consistent correlation between recombination and gene density may have implications for the evolution of recombination landscapes and whether the distribution of COs is optimal for the efficacy of genetic shuffling.

### Chromosome size and recombination rate

We showed that, for most species, the smallest chromosome had roughly one or two COs, independently of chromosome size. This is in agreement with the idea that CO assurance is a

ubiquitous regulation process among angiosperms [10]. Moreover, this constraint imposes a kind of basal recombination rate for each species, on the order of $50/S_c$ cM/Mb, where $S_c$ is the size of the lowest chromosome in Mb. Regardless of the genome size (which ranges three orders of magnitude or more), the number of COs remains relatively stable amongst species, most probably under the joint influence of CO assurance, interference and homeostasis [4,9,11]. As a result, averaged recombination rates are negatively correlated with chromosome lengths, as already known in plants [2,21].

However, there is no universal relationship between the absolute size of a chromosome and its mean recombination rate. Although the average recombination rate of a species is well predicted by its average chromosome size, the recombination rates of each chromosome separately are not well predicted by their absolute chromosome size. Instead, variation within species is much better explained by the relative chromosome size, and surprisingly, this relationship seems to be roughly the same among species (see Figs 1 and 2). This suggests that CO interference is proportional to the relative size of the chromosome, as it has been empirically observed in some plants [32]. Although it is not clear yet which interference distance unit is the most relevant, genomic distances (in Mb) are excluded in most models of interference in favour of genetic distances (cM) [33] or, more likely, the length of the synaptonemal complex in micrometres [5,34–36]. Both scales match our observation of a relative size effect. Within species, genetic maps increase with chromosome size, but among species they are uncorrelated and far less variable than genome sizes, which makes the relative chromosome size the main determinant of recombination rate variations among species. Similarly, physical sizes (in micrometres) at meiosis do not seem to scale with genome size, as chromosomal organization (nucleosomes, chromatin loops) strongly reduces the variation that could be expected given the genome size [9].

## Recombination patterns along chromosomes

We observed a global trend towards higher recombination rates in sub-distal regions [2,29]. The distal bias increased with chromosome length, in agreement with the conclusions of Haenel et al. [2], although our methods differ in resolution. We analysed species and chromosomes separately whereas Haenel et al. [2] used averages over the different patterns, thereby masking chromosome- and species-specific particularities. For example, they did not detect the sub-distal pattern neither unclassified exceptions, whereas they seem common among species (16 and 7 species respectively). So far, little is known about the mechanisms that could explain the link between the distal bias and chromosome length. Even if models of CO interference yield similar patterns [37,38], the conceptual model of Haenel et al. [2] is still the only one to explicitly consider chromosome length. The telomere effect is thought to act at a broad chromosome scale over long genomic distance. The decision of double strand breaks (DSBs) to engage in the CO pathway is made early on during meiosis and the early chromosome pairing beginning in telomeres is thought to favour distal COs [39–41]. In barley, when the relative timing of the first stages of the meiotic program was shortened, COs were redistributed towards proximal regions [40], as later observed in wheat [42].

Haenel et al. [2] proposed that distance to the telomere is driving CO positioning, and therefore it should produce a symmetrical U-shaped pattern along chromosomes. However, a formal test showed that this model was too simple and that centromeres also played a role in the distribution of COs between chromosome arms. The best model (M3: 'telomere + centromere + one CO per chromosome') that we have proposed suggests that centromeres do not only have a local effect but also influence the symmetry of recombination landscapes over long distance, though a large proportion of our sample is metacentric, which might limit the

detection of an effect. The local suppression of COs in centromeric regions is well known and largely conserved among species and seems a strong constitutive feature restricted to a short centromeric region, basically the kinetochore [43,44]. But the extent of the pericentromeric region varies drastically, most probably under the influence of DNA methylation, chromatin accessibility or RNA interference [14,43,45,46]. However, how centromeres (especially non-metacentric ones) may affect CO distribution at larger scales still needs to be determined.

## Diversity of patterns among species

In addition to the role of centromeres, we also observed that the distal model is not found in all plants. Instead, we observed at least two different crossover patterns among plant species (34 with the distal model and 16 with the sub-distal model), while seven species remain unclassified, which is at the limit of our visual classification. Globally, the distal pattern seems to occur more often in larger chromosomes, but our data lack species with giant genomes, which are not rare in plants [27]. Astonishingly, a low-density genetic map in *Allium* showed higher recombination rates in the proximal regions, which is opposite to the major trend we found [47]. Genera with giant genomes such as *Lilium* or *Allium* would have been valuable assets in our dataset, but the actual genomic and linkage data are relatively incomplete [48,49].

The occurrence of various recombination patterns is in agreement with what is known of the timing of meiosis and heterochiasmy (the fact that male and female meiosis have different CO patterns). Despite the strong conservation of the main meiotic mechanism in plants, differences in the balance between key components may produce distinct CO patterns [1,13,20,40]. For example, the ZYP1 and ASY1 proteins have antagonistic effects on the formation of the synaptonemal complex in plants [50]. In barley and wheat, linearization of the chromosome axis triggered by ZYP1 is gradual along the chromosome and initiated in distal regions, forming the telomere bouquet where early DSBs form [40,42]. In contrast, chromosome axes are formed at a similar time in *Arabidopsis thaliana* and chromosomes are gradually enriched in ASY1 from the telomeres to the centromeres; a gene-dosage component favours synapsis and ultimately COs towards the proximal regions [50]. It appears that the timing of the meiotic programme is important for the distal bias, as it involves changes in the relative contribution of each meiotic component that could explain the re-localization of COs [40,50]. Therefore, the different patterns we observed may be explained by the different balance and timing of the expression of shared key regulators of CO patterning such as ZYP1 and ASY1 [20]. It is interesting to note that this is also true for mechanistic models of interference. Zhang et al. [38] assessed that the 'beam-film' model is able to fit both CO patterns, regardless whether the tips of the chromosomes have an effect on interference or not, i.e. clamping. If clamping is assumed, the model predicts that mechanical stress culminates in the extremities of the chromosome leading to high CO rates at the periphery where it is released first. In contrast, when clamping is limited, mechanical stress is released in the tips of the chromosome and COs occur further from the tips, until a threshold of mechanical stress is reached. The observed sub-distal pattern fits these predictions.

The two patterns of recombination we described here can also be observed in opposite sexes within the same plant species [34,51,52]. Marked heterochiasmy variations between species, a feature shared among plants and animals, could influence the resulting sex-averaged recombination landscape [52]. The sex-averaged telomere effect can be thought of as the product of two independent sex-specific landscapes although it is not clear how sex-specific maps ultimately contribute to the sex-averaged one [53,54]. Recombination is usually biased towards the tips of the chromosome in male recombination maps, but is more evenly distributed in female maps in the few plant species with available data [52]. In *Arabidopsis thaliana*, male

meiosis has higher CO rates within the tips of the chromosome, as it has been observed in other species with large chromosomes, whereas female meiosis is more homogeneously distributed, with the lowest rates found in the distal regions [34]. Shorter chromosome axes in *A. thaliana* female meiosis could induce fewer DSBs and class II non-interfering COs [36]. Conversely, in maize, the distal bias is similar in both sexes, despite higher CO rates for females [55]. Heterochiasmy is not universal in plants [56], and we suggest that the variation in recombination landscapes could also result from variation in heterochiasmy among species, as it has been suggested for broad-scale differences in recombination landscapes between *A. thaliana* and its relative *A. arenosa* [51]. This hypothesis should be tested further as more sex-specific genetic maps become available.

## Recombination landscapes, gene density and genetic shuffling

We observed a strong convergence between CO patterns and gene density patterns. Interestingly, we found the same correlation in species with atypical chromosomes. For example, *Camellia sinensis* and *Nelumbo nucifera* have large genomes with homogenous recombination landscapes, and a recent annotation of the *Nelumbo nucifera* genome showed that genes are also evenly distributed along chromosomes at a broad scale [57], similar to *Camellia sinensis* [58]. In wheat and rye, the analysis of the effect of chromosome rearrangement on recombination also suggests that CO localization is more locus-specific than location-specific: after inversions of distal and interstitial segments, COs were relocated to the new position on the distal segment [59,60]. Overall, the parallel between gene density and recombination landscapes, confirmed by these two exceptions, is in agreement with the preferential occurrence of COs in gene regulatory sequences [14–16], and suggests that this may be a general pattern shared among angiosperms. Thus, gene distribution along chromosomes could be a main driver of recombination landscapes simply by determining where COs may preferentially occur. It should be noted that since the gene number is usually positively correlated with chromosome size within a species but is roughly independent of genome size among species, this hypothesis also matches with the relative-size effect discussed above.

However, gene density and recombination rates are both correlated with many other genomic features, such as transposable elements [25,61]. The accumulation of TEs in low recombining regions would progressively decrease gene density in the region, and would eventually result in a positive correlation between gene density and recombination. However, the correlation of recombination rates with TEs is not always clear and different TE families have opposite correlations [25,62]. On the one hand, gene density could directly determine recombination landscapes, leading to the accumulation of TEs in gene-poor regions, which would be amplified by a positive feedback loop. On the other hand, recombination could be targeted to gene-rich regions to avoid the deleterious effects of ectopic recombination between TEs [25]. Recombination, gene density and TEs could thus co-evolve and causal mechanisms of these multiple interactions still need to be clarified [25]. The use of fine scale recombination maps (using very large mapping populations or LD maps) should help identifying the respective role of genic regions (especially the role of promoters) and transposable elements (or other genomic features).

Irrespective of the underlying mechanism, our finding implies that the CO distribution ultimately scales with the gene distribution. Therefore, in most species, COs have a more even distribution between genes than between random genomic locations (Fig 10), which may have important evolutionary implications such as homogenizing the probability of two random genes to recombine, especially for large genomes that exhibit the strongest difference in genetic shuffling between genes and between genomic locations (Fig 11). Therefore, CO patterning

(and not only the global CO rate) could be under selection not only for its direct effect on the functioning of meiosis but also for its indirect effects on selection efficacy [9]. Recombination decreases linkage disequilibrium and negative interferences between adjacent loci (e.g. Hill-Robertson Interference), and thus locally increases the efficacy of selection. Functional sites are targets for selection [63] and we found higher recombination rates in functional regions, meaning that only a few genes are ultimately excluded from the benefits of recombination, even under the most pronounced distal bias.

Higher recombination rates in gene-rich regions could provide a satisfying explanation as to why the distal bias is maintained among species despite its theoretical lack of efficacy for genetic shuffling [31]. The association between CO hotspots and gene regulatory sequences is mechanistically driven by chromatin accessibility, but it does not exclude the evolution of the mechanism itself towards the benefits of recombining more in gene-rich regions [54]. However, slight variations in genetic shuffling caused by the non-random distribution of COs are less likely to be under strong selection compared to stabilizing selection on molecular constraints for chromosome pairing and segregation [64], although interference is sometimes likely to evolve towards relaxed physical constraints [9]. In addition, the intra-chromosomal component of the genetic shuffling is a small contributor to the genome-wide shuffling rate, as a major part is due to independent assortment among chromosomes [31]. Our estimates for the chromosomal genetic shuffling do not reach the theoretical optimal value of 0.5. The pattern is not absolute, and a fraction of genes remains in low recombining regions. In grass species, up to 30% of genes are found in recombination deserts and are not subject to efficient selection (e.g. [65]). Finally, it is still an open question as to whether this global distribution of COs in gene regulatory sequences is advantageous for the genetic diversity and adaptive potential of a species [66].

## Conclusion

Our comparative study only demonstrates correlations, and not mechanisms, but helps to understand the diversity and determinants of recombination landscapes in flowering plants. Our results partly confirm previous studies based on fewer species [2,4,21] while bringing new insights that alter previous conclusions thanks to a detailed analysis at the species and chromosome levels. Two main and distinct CO patterns emerge across a large set of flowering plant species; it seems likely that chromosome structure (length, centromere) and gene densities are the major drivers of these patterns, and the interactions between them raise questions about the evolution of complex genomic patterns at the chromosome scale [29,67]. The new large and curated dataset we provide in the present work should be useful for addressing such questions and testing future evolutionary hypotheses regarding the role of recombination in genome architecture, and we hope that many new species will be added, especially thanks to the increasing number of fully-sequenced plant genomes. We also encourage experimentalists to quantify separately male and female recombination landscapes, as potential sex differences could bring important insight on the evolution of recombination patterns [52]. Comparing recombination maps among closely related species, or even among population of a same species, would also help understanding how recombination landscapes evolve.

## Materials and methods

### Data preparation

To build recombination maps, we combined genetic and genomic maps in angiosperms that had already been published in the literature. We conducted a literature search to collect sex-averaged genetic maps estimated on pedigree data–with markers positions in centiMorgans (cM). The keywords used were 'genetic map', 'linkage map', 'genome assembly', 'plants' and

'angiosperms', combined with 'high-density' or 'saturated' in order to target genetic maps with a large number of markers and progenies. Additionally, we carried out searches within public genomic databases to find publicly available genetic maps. Only species with a reference genome assembly at a chromosome level were included in our study (a complete list of genetic maps with the associated metadata is given in S1 and S2 Tables). As much as possible, genomic positions along the chromosome (Mb) were estimated by blasting marker sequences on the most recent genome assembly (otherwise genomic positions were those of the original publication). Genome assemblies with annotation files at a chromosome-scale were downloaded from NCBI (https://www.ncbi.nlm.nih.gov/) or public databases. Marker sequences were blasted with 'blastn' and a 90% identity cutoff. Markers were anchored to the genomic position of the best hit. When the sequence was a pair of primers, the mapped genomic position was the best hit between pairs of positions showing a short distance between the forward and reverse primer ($< 200$ bp). In a few exceptions (see S1 Table), genomic positions were mapped on a close congeneric species genome and the genomic map was kept if there was good collinearity between the genetic and genomic positions. Chromosomes were numbered as per the reference genome assembly. When marker sequences were not available, we kept the genomic positions published with the genetic map. The total genomic length was estimated by the length of the chromosome sequence in the genome assembly. The total genetic length was corrected using Hall and Willis's method [30] which accounts for undetected events of recombination in distal regions by adding *2s* to the length of each linkage group (where *s* is the average marker spacing in the group).

We selected genetic and genomic maps after stringent filtering and corrections, using custom scripts available in a public Github repository (https://github.com/ThomasBrazier/diversity-determinants-recombination-landscapes-flowering-plants.git). We assumed that markers must follow a monotone increasing function when plotting genetic distances as a function of genomic distances in a chromosome (i.e. the Marey map) and collinearity between the genetic map and the reference genome was required to keep a Marey map. If necessary, genetic maps were reoriented so that the Marey map function is increasing (i.e. genetic distances read in the opposite direction). In a first step, Marey maps with fewer than 50 markers per chromosome were removed, although a few exceptions were visually validated (maps with ~30 markers). Marey maps with more than 10% of the total genomic map length missing at one end of the chromosome were removed. Marey maps with obvious artefacts and assembly mismatches (e.g. lack of collinearity, large inversions, large gaps) were removed. Markers clearly outside the global trend of the Marey map (e.g. large genetic/genomic distance from the global cloud of markers or from the interpolated Marey function, no other marker in a close neighbourhood) were visually filtered out, and multiple iterations of filtering/interpolation helped to refine outlier removal. The Marey map approach is a graphical method, so figures were systematically produced at each step as a way to evaluate the results of the filtering and corrections. Finally, when multiple datasets were available for the same species, we selected the dataset with the highest marker density–in addition to visual validation–to maintain a balanced sampling and avoid pseudo-replicates of the same chromosome.

## Estimates of local recombination rates

Local recombination rates along the chromosome were estimated with custom scripts following the Marey map approach, as described in the MareyMap R package [68]. The mathematical function of the Marey map was interpolated with a two-degree polynomial loess regression. Each span smoothing parameter was calibrated by 1,000 iterations of hold-out partitioning (random sampling of markers between two subsets; 2/3 for training and 1/3 for testing) with

the Mean Squared Error of the loess regression as a goodness-of-fit criterion. The possible span ranged from 0.2 to 0.5 and was visually adjusted for certain maps. This validation procedure to automatically adjust the smoothing allowed avoiding overfitting and underfitting issues. The local recombination rate was the derivative of the interpolated smoothed function in fixed 100 kb and 1 Mb non-overlapping windows. Negative estimates were not possible as we assumed a monotonously increasing function and negative recombination rates were set to zero. The 95% confidence intervals of the recombination rates were estimated by 1,000 bootstrap replicates of the markers to evaluate the sensitivity of our estimates to outliers and noisy data. Recombination landscapes with large confidence interval were discarded. The quality of the estimates was checked using the correlation between the 100 kb and 1 Mb windows. In addition, to check for a bias in our estimates (e.g. inflating the chromosome recombination rate), we assessed the differences between the genome wide recombination rate (obtained by dividing the genetic map length by the genome length) and the average estimate per chromosome (the mean of recombination rates in windows of 100 kb). Both values are extremely correlated (Spearman's Rho = 0.99, p < 0.001 and slope = 1).

## The distribution of CO along chromosomes

The spatial structure of recombination landscapes across species and chromosomes is a major feature of recombination landscapes. We divided the Marey map in $k$ segments of equal genomic size (Mb) and then calculated the relative genetic size (cM) of each segment. Under the null model (i.e. random recombination), one expects $k$ segments of equal genetic size $1/k$. The relative recombination rate in the segment $i$ was estimated by the log-ratio of the observed genetic size (i.e. genetic size of segment $i$) divided by the expected genetic size (i.e. fixed to total genetic size / $k$ by the model), as in the following equation.

$$relative\ recombination\ rate = log_{10} \frac{genetic_i}{genetic_{total}/k}$$

Given the observation that most recombination landscapes are broken down into at least three segments [69], we arbitrarily chose a number of segments $k = 10$ to reach a good resolution (a larger $k$ did not show any qualitative differences).

## Crossover patterns and the periphery-bias ratio

We investigated the spatial bias towards distal regions of the chromosome in the distribution of recombination by estimating recombination rates as a function of relative distances to the telomere (i.e. distance to the nearest chromosome end). Chromosomes were split by their midpoint and only one side was randomly sampled for each chromosome to avoid pseudo-replicates and the averaging of two potentially contrasting patterns on opposite arms. The relative distance to the telomere was the distance to the telomere divided by total chromosome size, then divided into 20 bins of equal relative distances. A periphery-bias ratio metric similar to the one presented in Haenel et al. [2] was estimated to measure the strength of the distal bias. We divided the recombination rates in the tip of the chromosome (10% on each side of the chromosome, and one randomly sampled tip) by the mean recombination rate of the whole chromosome. We investigated the sensitivity of this periphery-bias ratio to the sampling scale by calculating the ratio for many distal region sizes (S13 Fig).

## Testing centromere or telomere effects

We searched the literature for centromeric indices (ratio of the short arm length divided by the total chromosome length) established by cytological measures. When we had no

information about the correct orientation of the chromosome (short arm/long arm), the centromeric index was oriented to match the region with the lowest recombination rate of the whole chromosome (i.e. putative centromere). To determine if telomeres and centromeres play a significant role in CO patterning, we fitted empirical CO distributions to three theoretical models of CO distribution. In the following equations, $d(x)$ is the relative genetic distance at the relative genomic position $x$, and $a$ is a coefficient corresponding to the excess of COs per genomic distance. Under the strict 'telomere' model (1), we assumed that only telomeres played a role in CO distribution, i.e. an equal distribution of COs on both sides of the chromosome (i.e. $d(1/2) = d(1)-d(1/2)$, such that $\frac{d(1/2)}{d(1)} = 0.5$. The 'telomere + centromere + one mandatory CO per arm' model (2) assumed at least one CO per chromosome arm and a relative genetic distance of each chromosome arm proportional to its relative genomic size, corresponding to the role of centromere position, denoted $d(c)$. We have $d(c) = 50+a{\times}c$ and $d(1)-d(c) = 50+a{\times}(1-c)$, such that $\frac{d(c)-50}{d(1)-100} = c$. Lastly, the 'telomere + centromere + one CO per chromosome' model (3) assumed at least one CO per chromosome and a relative genetic distance within the chromosome proportional to its relative genomic distance. We have $d(c) = c{\times}50 +a{\times}c$ and $d(1)-d(c) = (1-c){\times}50+a{\times}(1-c)$, such that $\frac{d(c)}{d(1)} = c$. The three competing models were compared with a linear regression between empirical and theoretical values, based on the adjusted $R^2$ and AIC-BIC criteria among chromosomes. The number of species supporting each model was calculated based on the adjusted $R^2$ within species, for all species with at least five chromosomes.

## Gene density

We retrieved genome annotations ('gff' files) for genes, coding sequences and exon positions, preferentially from NCBI and otherwise from public databases (41 species). We estimated gene counts in 100 kb windows for recombination maps by counting the number of genes with a starting position falling inside the window. For each gene count, we estimated the species mean recombination rate and its confidence interval at 95% by 1,000 bootstrap replicates (chromosomes pooled per species). Most species had rarely more than 20 genes over a 100 kb span and variance dramatically increased in the upper range of the gene counts, and therefore we pruned gene counts over 20 for graphical representation and statistical analyses.

## Genetic shuffling

To assess the efficiency of the recombination between chromosomes and species, we calculated the measure of intra-chromosomal genetic shuffling described by Veller et al. [31]. To have even sampling along the chromosome, genetic positions (cM) of 1,000 pseudo-markers evenly distributed along genomic distances (Mb) were interpolated using a loess regression on each Marey map, following the same smoothing and interpolation procedure as for the estimation of the recombination rates. The chromosomal genetic shuffling $\bar{r}_{intra}$ were calculated as per the intra-chromosomal component of the equation 10 presented in Veller et al. [31]. For a single chromosome,

$$\bar{r}_{intra} = \sum_{i<j}\left(r_{ij}/\binom{\Lambda}{2}\right)$$

where $\Lambda$ is the total number of loci, $\binom{\Lambda}{2} = \Lambda(\Lambda-1)/2$ and $r_{ij}$ is the rate of shuffling for the locus pair $(i, j)$. For the intra-chromosomal component $\bar{r}_{intra}$, the pairwise shuffling rate was only calculated for linked sites, i.e. loci on the same chromosome. This pairwise shuffling rate

was estimated by the recombination fraction between loci *i* and *j*. Recombination fractions were directly calculated from Haldane or Kosambi genetic distances between loci by applying a reverse Haldane function (1) or reverse Kosambi function (2), depending on the mapping function originally used for the given genetic map.

$$r_{ij} = \frac{1}{2}\left(1 - e^{-2d_{ij}/100}\right) \tag{1}$$

$$r_{ij} = \frac{1}{2}tanh\left(2d_{ij}/100\right) \tag{2}$$

We also estimated marker positions in gene distances instead of genomic distances (Mb) to investigate the influence of the non-random distribution of genes on the recombination landscape. Gene distances were the cumulative number of genes along the chromosome at a given marker's position. Splicing variants and overlapping genes were counted as a single gene. The genetic shuffling was re-estimated with gene distances instead of genomic distances to consider a genetic shuffling based on the gene distribution, as suggested by Veller et al. [31]. To compare the departure from a random distribution along the chromosome among both types of distances (i.e. genomic and genes), we calculated the Root Mean Square Error (RMSE) of each Marey map and for both distances. To assess if the distribution of genes influenced the heterogeneity of recombination landscapes, the type of distance with the lower RMSE was considered as the more homogeneous landscape. However, this measure for gene distances is sensitive to annotation errors and artefacts. False negatives are therefore expected (when Marey maps were assessed as more homogeneous in genomic distances while the inverse is true) and this classification remains conservative.

## Statistical analyses

All statistical analyses were performed with R version 4.0.4 [70]. We assessed statistical relationships with the non-parametric Spearman's rank correlation and regression models. Linear Models were used for regressions with species data since we did not detect a phylogenetic effect. The structure in the chromosome dataset was accounted for by Linear Mixed Models (LMER) implemented in the 'lme4' R package [71] and the phylogenetic structure was tested by fitting the Phylogenetic Generalized Linear Mixed Model (PGLMM) of the 'phyr' R package [72]. The phylogenetic time-calibrated supertree used for the covariance matrix was retrieved from the publicly available phylogeny constructed by Smith and Brown [73]. Marginal and conditional $R^2$ values for LMER were estimated with the 'MuMIn' R package [74]. Significance of the model parameters was tested with the 'lmerTest' R package [75]. We selected the model based on AIC/BIC criteria and diagnostic plots. Reliability and stability of the various models were assessed by checking quantile-quantile plots for the normality of residuals and residuals plotted as a function of fitted values for homoscedasticity. Model quality was checked by the comparison of predicted and observed values. Given the skewed nature of some distributions, we used logarithm (base 10) transformations when appropriate. For comparison between species, statistics were standardized (i.e. by subtracting the mean and dividing by standard deviation). Mean statistics and 95% confidence intervals were estimated by 1,000 bootstrap replicates.

## Supporting information

**S1 Fig. Markers positions in genetic distance (cM) as a function of genomic distance (Mb), namely Mary maps, for each chromosome included in the dataset (n = 665 chromosomes).**

The black vertical line is the centromere position estimated by cytological measures, when available in the literature.
(PDF)

**S2 Fig. Recombination landscapes for each chromosome included in the dataset (n = 665 chromosomes).** Recombination rate (cM/Mb) estimated in windows of 100kb along genomic distances (Mb). Confidence interval at 95% (grey ribbon) estimated by 1,000 bootstraps of loci. The black vertical line is the centromere position estimated by cytological measures, when available in the literature.
(PDF)

**S3 Fig. Dataset quality for the 57 species.** The averaged linkage map length (total linkage map length divided by the number of chromosomes, cM) is not correlated with (A) the number of markers (linkage map length ~ log10(number of markers), adjusted $R^2$ = 0.04, p = 0.11), (B) marker density (linkage map length ~ marker density, adjusted $R^2$ = -0.018, p = 0.90) and (C) the progeny size (linkage map length ~ progeny size, adjusted $R^2$ = 0.022, p < 0.32). Regression lines with 95% parametric confidence interval estimated with ggplot2.
(TIF)

**S4 Fig. Phylogenetic tree of species in our dataset (n = 57), annotated with mean recombination rate (cM/Mb) and mean chromosome size (Mb).** The supertree was retrieved from the publicly available phylogeny constructed by Smith and Brown (Smith & Brown, 2018).
(TIF)

**S5 Fig. Slopes of the linear regression within species (linkage map length ~ chromosome size) as a function of the species mean genomic chromosome size (Mb).**
(TIF)

**S6 Fig. The negative correlation (Spearman's Rho coefficient) between recombination rates (cM/Mb) and the distance to the nearest telomere is stronger for species with a larger chromosome size (n = 57).** The linear regression line and its parametric 95% confidence interval were estimated in ggplot2. The inset presents the distribution of Spearman's Rho coefficients for chromosomes (n = 665 chromosomes). The mean correlation and its 95% confidence interval (black solid and dashed lines) were estimated by 1,000 bootstraps. The red vertical line is for a null correlation.
(TIF)

**S7 Fig. Standardized recombination rate (cM/Mb) as a function of the relative distance (Mb) from the telomere along the chromosome (physical distances expressed in 20 bins).** Chromosomes were split in halves, a relative distance of 0.5 being the centre of the chromosome, and only one side was randomly sampled to avoid averaging patterns. Then, chromosomes were pooled per species. Each colour is a species. A loess regression was estimated for each species. Species presented in four plots for clarity.
(TIF)

**S8 Fig. Standardized gene count as a function of the relative distance (Mb) from the telomere along the chromosome (physical distances expressed in 20 bins).** Chromosomes were split in halves, a relative distance of 0.5 being the centre of the chromosome, and only one side was randomly sampled to avoid averaging patterns. Then, chromosomes were pooled per species. Each colour is a species. A loess regression was estimated for each species. Species presented in four plots for clarity.
(TIF)

**S9 Fig. The genetic shuffling $\bar{r}_{intra}$ increases with the size of the genetic map (cM).** Linear mixed regression with a species random effect and its 95% confidence interval estimated by ggplot2 (black line and grey ribbon). Each colour is a species. A linear regression was estimated for each species.
(TIF)

**S10 Fig. The genetic shuffling $\bar{r}_{intra}$ decreases with the periphery-bias ratio. Linear mixed regression with a species random effect and its 95% confidence interval estimated by ggplot2 (black line and grey ribbon).** Each colour is a species. A linear regression was estimated for each species.
(TIF)

**S11 Fig. Gene count in windows of 100kb along genomic distances (Mb) for each chromosome with gene annotations (n = 480 chromosomes).** Recombination rate (cM/Mb) estimated in windows of 100kb. Loess regression of gene count along the chromosome in blue line with parametric confidence interval at 95% in grey.
(PDF)

**S12 Fig. Marey maps with genomic distances (black points) and gene distances (gray points). Markers positions in genetic distance (cM) as a function of the relative physical distance (either Mb of cumulative number of genes) for each chromosome with gene annotations (n = 480 chromosomes).** The black dashed line is a theoretical uniform distribution of markers. The black vertical line is the centromere position estimated by cytological measures, when available in the literature.
(PDF)

**S13 Fig. Sensitivity of the periphery-bias ratio to the size of the sampled distal region (i.e. number of bins sampled at the tips).** The periphery-bias ratio was estimated for different numbers of bins sampled and always divided by the mean chromosomal recombination rate. Linear regression (black line) shows a decrease of the periphery-bias ratio as the number of bins increases, towards a ratio value of 1 (dashed line).
(TIF)

**S1 Table. Metadata for 665 recombination landscapes, with name of the dataset collected and literal name of the chromosome used in our study, chromosome name in annotation (gff), size of the genetic map (cM, raw and corrected by methods of Chakravarti et al. (1991) or Hal & Willis (2005)), size of the genomic sequence in genome assembly (Mb), number of markers, density of markers in cM and bp, progeny size, mean interval between markers in cM and bp, Gini index, span parameter of the loess function, type of mapping function (Haldane, Kosambi or none), accession of the reference genome used for markers genomic positions, link to data repository and doi reference of the study in which the genetic map was published.**
(XLSX)

**S2 Table. Flowering plant species included in the study, with authors, year and doi reference of the genetic map publication, and accession of the reference genome.**
(XLSX)

**S3 Table. Centromeric indexes estimated in cytological studies, with unit of measurement, mean and standard error of long and short chromosome arms, centromeric index (ratio of short arm length divided by total chromosome length), and doi reference to the original**

**study.**
(XLSX)

**S4 Table. Correlation between recombination landscapes estimated at two different genomic scales (1Mb and 100kb).** Spearman's Rho coefficient was estimated for each chromosome between recombination rates estimated directly in windows of 1Mb and the mean recombination rate of 100kb windows pooled together in 1Mb windows. Mean of the Spearman's Rho coefficient among chromosomes and proportion of significant p-values given for each species.
(XLSX)

**S5 Table. Selection of the regression model between LM, LMER and PGLMM which explains best the relationship between the mean recombination rate (cM/Mb) and the chromosome size (Mb), based on AIC and BIC criteria.**
(XLSX)

**S6 Table. Species averaged correlation between the averaged chromosome size (Mb) and the averaged periphery-bias ratio.** Mean of the Spearman's Rho coefficient among correlations at chromosome scale and proportion of significant p-values given for each species.
(XLSX)

**S7 Table. Chromosome correlation between the recombination rate (cM/Mb) and the relative distance to the telomere, with Spearman's Rho coefficient and p-value of the test per chromosome.**
(XLSX)

**S8 Table. Species averaged correlation between the recombination rate (cM/Mb) and the relative distance to the telomere.** Mean of the Spearman's Rho coefficient among correlations at chromosome scale and proportion of significant p-values given for each species.
(XLSX)

**S9 Table. Selection of the best model of crossover distribution for each species, based on Adjusted R-Squared between observed values and theoretical values predicted by the model.** The best model selected for each species is the one maximizing the Adjusted R-Squared.
(XLSX)

**S10 Table. Selection of the best model of crossover distribution for each species in a subset of chromosomes with at least 50cM on each chromosome arm, based on Adjusted R-Squared between observed values and theoretical values predicted by model.** The best model selected for each species is the one maximizing the Adjusted R-Squared.
(XLSX)

**S11 Table. Convergence between crossover patterns and gene patterns at a species scale.** For each species is given the type of crossover pattern, the type of gene count pattern, the difference RMSE(gene pattern)—RMSE(crossover pattern) which indicates how gene patterns are more/less homogeneous than crossover patterns, the homogenization effect of gene patterns (more/less), the difference genetic shuffling(gene pattern)—genetic shuffling(crossover pattern) and the averaged chromosome size (Mb).
(XLSX)

**S1 Data. References for linkage map data included in this study.**
(PDF)

## Acknowledgments

We thank Eric Jenczewski, Laurent Duret, Anne-Marie Chèvre, Eric Petit, Armel Salmon and Bruno Raquillet for precious comments on the results and manuscript. We thank all the people that provided us genetic data.

## Author Contributions

**Conceptualization:** Sylvain Glémin.

**Data curation:** Thomas Brazier.

**Formal analysis:** Thomas Brazier.

**Funding acquisition:** Sylvain Glémin.

**Methodology:** Thomas Brazier.

**Supervision:** Sylvain Glémin.

**Validation:** Sylvain Glémin.

**Writing – original draft:** Thomas Brazier.

**Writing – review & editing:** Thomas Brazier, Sylvain Glémin.

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
