## [Decision Letter · Decision Letter 0]

26 Apr 2022

Dear Dr Brazier,

Thank you very much for submitting your Research Article entitled 'Diversity and determinants of recombination landscapes in flowering plants' to PLOS Genetics.

The manuscript was fully evaluated at the editorial level and by three independent peer reviewers. The reviewers appreciated the attention to an important problem, but raised some substantial concerns about the current manuscript. Based on the reviews, we will not be able to accept this version of the manuscript, but we would be willing to review a much-revised version. We cannot, of course, promise publication at that time.

If you decide to revise the manuscript for further consideration at PLOS Genetics, please aim to resubmit within the next 60 days, unless it will take extra time to address the concerns of the reviewers, in which case we would appreciate an expected resubmission date by email to plosgenetics@plos.org.

[LINK]

We are sorry that we cannot be more positive about your manuscript at this stage. Please do not hesitate to contact us if you have any concerns or questions.

Yours sincerely,

Ian Henderson

Associate Editor

PLOS Genetics

Kirsten Bomblies

Section Editor: Evolution

PLOS Genetics

Reviewer's Responses to Questions

**Comments to the Authors:**

Reviewer #1: It is now possible to analyse genetic and physical maps of organisms, as the literature now contains suitable data from multiple species. This manuscript analyses such data from larger numbers of plant species than previous studies (55 species, 5-26 chromosomes per species), and describes results for the broad-scale recombination landscapes. However, it does not actually use the analyses to ask interesting questions, which I had hope to see. Some questions might include the following

Do chromosome arms have an obligate crossover?

How often do chromosome arms have multiple crossovers, versus a single one (as I believe is the case in C. elegans)?

Do related species differ (this is an important question, as it relates to the question of whether genetic recombination is sometimes selectively favoured, leading to higher crossover numbers than required for correct segregation, and for repair mechanisms to occur). This question is discussed near the very end of the text, but is not mentioned as a question earlier, making it appear that the ms is entirely descriptive. The ms does not seem to mention that some of the species studied are close relatives, and that this can be helpful in studying such questions.

How large are pericentromeric regions with low recombination rates in plants, and how much do they differ between related species?

Do selfers have higher recombination rates per physical length of chromosome than closely related outcrossers?

Are recombination rates the same in male versus female meiosis? This is finally mentioned in line 574, but it is not made clear until then that the data analysed are sex-averaged rates.

Instead, the ms presents rather dull statistical analyses. The results have value, but they appear mainly to confirm findings that were already well established, and the ms does not make very clear what new findings now emerge, or show what we can now understand from the results that was not already known. More than once in the text “new insights” are claimed, but it is difficult to find them, partly because of the length of the text, which is also long-winded and repetitive in several places. These problems could be ameliorated by outlining in the Introduction what questions the authors set out to study. As written, this section gives the impression that their aims were purely descriptive, which is not an encouragement to read the text. The ms also tries to interest the reader by making claims to novelty, rather than describing some interesting questions. For example, I feel that it is too strong to say that “the broad diversity of recombination landscapes among plants has rarely been investigated… and the diversity of the resulting landscapes among species and chromosomes still need[s] to be assessed“, although a formal comparative genomic approach may be new and valuable. A further value from analysing more species is that exceptions to accepted generalisations may be detected, and this study did produce a few examples of such exceptions. Overall, I doubt that readers need a length Introduction to tell them that recombination patterns are interesting in relation to evolution, including evolution of patterns in genomes, such as regions with different repetitive sequence density, and consequently gene density, and with differences in GC content. A shorter Introduction could give a better idea of what is new from this study.

At least several of the conclusions are just confirmations of what was already known. The following examples illustrate this problem, and my comments also include some other issues for some of them (a recurring problem throughout the text is poor writing, including long-winded writing that makes the meaning hard to understand, and I provide some examples in my ‘Minor comments’ below, but these are still important comments that require revisions of the text, including a suggestion that some species may have too little information to be used. It would be helpful to show the numbers or markers mapped in Figure 4. In addition, if the numbers are small, presumably the total genetic map lengths are unreliable, and it is not explained prominently whether any attempt was made to check for this problem.

1. “We observed that the bias towards the periphery was not ubiquitous across species“ and “Only a subset of species, especially those with larger chromosomes, exhibited a clear bias”. These conclusions are quite similar to that of Haenel et al. (2018) that a distal bias is “universal for chromosomes larger than 30 Mb” (note the incorrect English “concluded to a distal bias”). The main advance seems to be that this study finds that Nelumbo nucifera and Camellia sinensis are exceptions to this pattern, with the highest recombination rates found in the middle of their chromosomes.

The result is described in a rather unhelpful manner, without taking chromosomes morphology into account. The text states that, for larger chromosomes, crossovers tend to occur (not “accumulate”) at the ends of chromosome, while the central regions have less. However, this would be correct only for metacentrics, and the centres of chromosome presumably means centromeric and pericentromeric regions, but this is not made clear. It is also not made clear that these are completely recombination-free regions.

The extent of a larger pericentromeric region (meaning, the extent of the wider region surrounding or adjacent to the centromere) is known to vary greatly between species, but it is not well described in the ms, and only examples are shown, with rather subjective criteria to define the different regions. It would, in principle, be possible to define them less subjectively, though this might not be easy. At least, it would be good to mention whether this was attempted. A further problem is that regions are shown in figures, rather than tables giving estimates of genome region sizes and recombination rates, and as relative sizes are often used, it is difficult to understand what sizes of pericentromeric regions (for example) are found in plants.

It is also not a new discovery that low recombination regions tend to have low gene density. The Discussion acknowledges this, but it is strange to first describe this as if it is a new result, only to later mention that it is not. If the Introduction had laid out some questions, this could be avoided. Problems like this also make the text longer than necessary.

2. Recombination is unevenly distributed in genomes. Therefore one should not write that “We showed that” this is the case. Once can write “We confirmed that” (or something similar). This text also uses vague terminology “how genetic variation is shuffled during meiosis”, but the word recombination already exists, so it would be better to be precise. If at some point the meaning is gene conversion, this should be used. However, I think that the text mentions conversion only in passing, and it is not considered seriously.

In line 538, I am not sure why thw word “prediction” is used (In addition to the role of centromeres, we also observed a departure from the prediction that recombination rates should decrease with the distance to the tip of the chromosome, showing that the distal model is not generally found among plants). Is this really a prediction, or are you trying to say that you did not confirm the view that this pattern is shared by all plants? If so, references are needed to assertions that all plants share this pattern.

The Discussion section need not repeat so much of the results. It might also mention that recombination rates vary between individuals of the same species, including from the effects of rearrangements, especially inversions, so it would be good to mention that the data are currently often from just a single maternal and paternal parental individual of each species (for selfers, perhaps just a single parental individual). Hotspots should also be mentioned, if only to make clear that this study did not attempt to detect them.

474 It is proposed that in angiosperms crossovers may be initiated in gene regulatory sequences, and it is suggested that this “sheds new light on the evolution of recombination landscapes”, but without saying what new light is shed, other than this suggestion. The suggestion is not evaluated further, and I did not understand if it is a speculation, based on the correlation between recombination and gene density mentioned in this paragraph (or on some other observations). However, based on later text (line 613), I suspect that the intended meaning is that the results are consistent with such a proposal that was already published by others.

However, as the correlation must be strongly affected by the lower gene densities in genome regions with low recombination rates, which lead to accumulation of transposable elements and other repetitive sequences, it would seem difficult to disentangle this from the suggested mechanism. Line 628 states that “The positive association of COs and gene regulatory sequences, including fine-scale correlations, appears more robust”, which is too vague. It seems unlikely that the effect is stronger than the very marked and consistent effect of low recombination rates on repetitive sequence density (although of course different elements are involved in different cases).

Regions with high recombination rates may, however, allow patterns in crossover localisation to be detectable, and I believe that this has been studied, for example in maize (e.g. papers by Dooner and colleagues) and also in Mimulus guttatus (see the paper by Hellsten et al. cited above). Line 621 finally mentions the problem of other correlated factors. I think that the authors should revise their text so that it does not first set up an untestable idea and then mention that it is untestable. Instead, it will be preferable to set up some interesting questions early in the text, tell readers what is currently known, and then describe analyses that help understand things better than before.

Dooner, H., & He, L. (2008). Maize genome structure variation: Interplay between retrotransposon polymorphisms and genic recombination. Plant Cell, 20(2), 249-258. doi:10.1105/tpc.107.057596

Fengler, K., Allen, S. M., Li, B., & Rafalski, A. (2007). Distribution of genes, recombination, and repetitive elements in the maize genome. Crop Science, 47(Supplement), S-83-S-95.

Yao, H., Zhou, Q., Li, J., Smith, H., Yandeau, M., Nikolau, B. J., & Schnable, P. S. (2002). Molecular characterization of meiotic recombination across the 140-kb multigenic a1-sh2 interval of maize. Proceedings of the National Academy of Sciences of the USA, 99, 6157-6162.

Tenaillon, M. I., Sawkins, M. C., Anderson, L. K., Stack, S. M., Doebley, J. F., & Gaut, B. S. (2002). Patterns of diversity and recombination along chromosome 1 of maize (Zea mays ssp. mays L.). Genetics, 162, 1401-1413.

Another comment that applies throughout the text is that recent papers are cited for concepta and understanding that are not new. In such cases, the text should make clear that the citation is to a review paper. For example, the text gives the impression that Marand et al. (2019) discovered that gene density and recombination rates are both correlated with transposable elements (meaning densities of transposable elements). This has been known for a long time, and was reviewed in 1994 by Charlesworth et al. (Nature, 371, 215-220. doi:10.1038/371215a0).

In first mentioning heterochiasmy, it seems strange not to mention whether the papers cited refer to plants or just to studies in animal species. It is explained later that Melamed-Bessudo et al. (2016) showed that it is not universal in plants, but the text does not explain what the term might mean in plants, and that hermaphrodites may have different crossover patterns in male and female meiosis, so readers may be puzzled.

3. Similarly, I was surprised to read that “We were intrigued to notice that [within species]

the chromosome-wide recombination rate is proportional to the relative size of the chromosome”. I was under the impression that this was already known.

It is illustrated in Figure 2D, which shows the new results, which are potentially interesting, as they relate to the question of how often arms have multiple crossovers. This figure analyses the excess of crossovers, defined as the linkage map length minus the 50 cM expected if one crossover per arm is obligate), and shows that it correlates positively with the chromosome’ physical sizes divided by the average chromosome size for the species, which they term the “relative chromosome size”. Such an effect is not a new result.

However, as I understand it, an obligate crossover is expected on each arm. If so, the number of excess crossovers, in addition to this one, should be analysed per arm. Even if my recollection about this is incorrect, the text should make clear what is known from previous studies, and why the present study uses chromosome, not arm, lengths. Line 136 mentions that the centromeric index was known for the chromosomes of 37 species, but then it remains unclear how these data were used, and also whether results can be used from the species where no such data were available. Line 285 mentions that recombination rates were negatively correlated with the distance to the nearest telomere, which seems to suggest that metacentrics may have been analysed as such, but I could not see this clearly explained.

Line 300 states that that (in my wording) the centromere regions almost universally showed low recombination rates, but this is not completely clear in Figure 4, where large low recombination rate regions in several species, for example Vigna unguiculata, appear not to overlap the centromeres. If this is a real biological observation, the statement seems incorrect.

Given these possible problems with the data, I was not convinced of the value of the formal modelling analysis of the effect of the centromere in suppressing recombination, and the comparison with less simple models that suggest that telomeres may also affect patterns. Such effects are plausible, but I feel that some of these plant data do not add valuable and solid support.

Another weakness is the lack of any mention of differences between male and female meiosis, and another is the lack of any mention of outcrossing rates.

I wondered why these papers were not cited, or other papers about Arabidopsis lyrate or helleri, which may have genetic map information.

Hellsten, U., Wright, K. M., Jenkins, J., Shu, S., Yuan, Y., Wessler, S. R., . . . Rokhsar, D. S. (2013). Fine-scale variation in meiotic recombination in Mimulus inferred from population shotgun sequencing. Proceedings of the National Academy of Sciences of the United States of America, 110(48), 19478–19482. doi:10.1073/pnas.1319032110

Kawabe, A., Hansson, B., Forrest, A., Hagenblad, J., & Charlesworth, D. (2006). Comparative gene mapping in Arabidopsis lyrata chromosomes 6 and 7 and A. thaliana chromosome IV: evolutionary history, rearrangements and local recombination rates. Genetical Research, 88, 45-56.

Hansson, B., Kawabe, A., Preuss, S., Kuittinen, H., & Charlesworth, D. (2006). Comparative gene mapping in Arabidopsis lyrata chromosomes 1 and 2 and the corresponding A. thaliana chromosome 1: recombination rates, rearrangements and centromere location. Genetical Research, 87(2), 75-85. doi:10.1017/S0016672306008287

Minor problems with the English, or vague wording or unclear statements

1. In English, it should be “correlated with” (not “to”).

2. The word ‘drive’ should be avoided, as it is very vague. For example, the meaning is not clear in the phrase “Chromosome length drives the basal recombination rate for each species”

3. In line 182, it should read “regression lines for species with at least 5 chromosomes mapped, 5-26 chromosomes per species, 55 species).

4. Line 232 Genomic distances (Mb) were scaled between 0 and 1 (divided by chromosome size) to compare chromosomes with different sizes.

5. It is difficult to make out the meaning of the text starting in line 247. I think it means the following: “Each chromosome was divided in (it should read “into”) ten bins, each one 10th of the chromosome’s total physical size.” The relative recombination rate is the log-transformed ratio of the expected relative genetic length (one tenth, presumably of the total genetic length) divided by the observed relative genetic length of the bin (presumably meaning the proportion of the total genetic length represented by the physical region in question. Values below zero correspond to recombination rates lower than expected under a random distribution of crossovers across the physical chromosome. Also difficult to understand “Chromosome sizes (Mb) on the left correspond to each broken stick chromosome” — maybe it means “each chromosome”. Also (in line 244) “Relative recombination rates along the chromosome were estimated in ten bins using the broken stick model.

6. In English, one needs to say “divided into” (not “in”). Also “pooled into” (although this reads awkwardly in English, and line 140 might be better as “the Spearman rank correlation coefficient correlation between the values for 1 Mb windows and those for the 100 kb windows within them was ….”.

7. The work “linkage” in genetics means that the variants are linked. It should be distinguished from “linkage disequilibrium” (LD), which refers to associations between two or more liked variants. Line 57 should be corrected, as the text refers to the latter, but uses the former (“Recombination…. breaking the linkage between neighbouring sites and creating new genetic combinations”). The sites remain linked, but not in LD. The sentence is also confusing by adding “upon which selection can act”, because selection acts on single variants, and the authors are trying to say that new genetic combinations might be more (or less) favoured by selection than the non-recombinant combinations (in other words, the different variants may interact in their effect on fitness).

8. It is a sweeping statement to say that “Plant genomes contain large regions with suppressed recombination”, depending strongly on how many plants have good data on physical and genetic maps, so line 92 ought to mention the number on which this is based, and give readers at least a rough idea of what is meant by “large”. There is no need to add the obvious remark that this impacts genomic averages ( in addition “impact” is the wrong word, as the meaning is that it affects the average — of course the average depends on the values in all genome regions that are included in the data, so it is not worth saying explicitly).

9. Phrases that are unnecessary (such as “it seems that” in line 93, should be pruned out, so that the text is easier to read. There are quite a few such instances, and I do not comment on all of them. The beginning of the Results section, for example, could be written more briefly and clearly.

We retrieved publicly available data for linkage maps and genome assemblies, to obtain genetic map distances and physical distances. We used linkage maps with marker positions in chromosome-level genome assemblies (except for Capsella rubella, which had a high-quality scaffold-level assembly of pseudo-chromosomes). After filtering based on the marker numbers, densities, and genome coverage, and after filtering out the outlying markers (maybe meaning outlier markers by a criterion that needs to be explained), we produced 665 Marey maps (reference needed) for 57 species (2-26 chromosomes per species); marker numbers per chromosome (or perhaps the authors mean per species, in which case perhaps some species have too little information to be used) ranged from 31 to 49,483.

.

Reviewer #2: In this paper, the authors seek to decipher genomic patterns of recombination across a large (57 species) dataset of sequenced plant genomes coupled with genetic maps. Their meta-analyses lead to several novel observations.

I thoroughly enjoyed reading this manuscript, and I congratulate the authors on a really fine paper. It will be, in my view, a very welcome addition to the literature. In the surest sign of flattery, I’m a jealous that I did not think of doing such a neat analysis.

Accordingly, I have only minor comments that the authors may wish to address in revision. Most of the comments are very minor, indeed. They are offered both as an attempt to clarify the few areas of the text that I found difficult to digest and probably out of an abundance of enthusiasm for this work. I leave it to the authors to decide if my suggestions offer improvements or are better ignored…

Minor Comments:

- Line 48 – Unlike most of the rest of the paper, I found this sentence hard to read and digest. Reword, rework or shorten? Btw I’d use “in” instead of “to” (“in the production”)

- Line 79 – This last sentence of the paragraph is really indirect and therefore pretty tough to read. I’m not really sure what manipulations are being considered here… Rewrite?

- Line 100 – as a reader, I found that a better link between the two sentences on this line could have been helpful. Maybe something as simple as “Haenel et al. considered chrosomome length, found blah blah blah and suggested a simpler telomere-led model. That model included a universal bias…”

- Line 118 – I’d use “about” instead of “on”

- Line 125 – If this is reasonable, I’d love to see the filter characteristics hinted at here, even though there is a good description in the methods. That is something like “… marker density (at least 50 per chromosome), genome coverage (blah blah)”

- Line 701 – I’m a bit confused by the what was done when marker sequences were not available and also how many species fell into this category. I’m not concerned at all – this is a careful study – but it’d be nice understand better.

- Figure 2 – It might be helpful to have X-axis say “Mean chromosome size” where appropriate (e.g., Figure 2C and B). The legend is very clear, though.

- I love Figure 3. It blows my mind how consistent the patterns are between the dashed lines (genome wide) and an individual chromosome. It is bizarre and neat and thought provoking. It might be nice to report mean chromosome size (in the legend or in the figure), given that the species are ordered in that matter. It just makes me curious…

- Figure 4, since patterns seem to correlated with mean chromosome size, would it be worth adding that value after each species name? As a reader, it would help me to see the pattern and better digest the text from ~lines 213 to 227 and figure 5a, etc).

- Figure 5A – this may make the graph too crowded, but it’d be nice to be able to compare dots in 5A to figure 4. So, it’d be nice to have the dots labelled. If that is too much, the authors might want to consider labelling a few species (e.g., the six in figure 3 or some of the species mentioned in lines 220 to 227). Personally, I’d love to know what the outliers are in this graph!

- Lines 284 and following. It’d be nice to cite Figure 6A and Figure 6B separately after the word descriptions of the patterns.

- Line 296. I could not figure out what the “species correlation” referred to. Sorry if I missed this, but it’s worth another look to be sure it is clear.

- I’m kind of shocked that M3 is favored over M2, as isn’t one CO per arm necessary for mechanism? Hence, I’d a priori predict M2 > M3. I don’t think this contrast is explicitly discussed in the Discussion (e.g., lines 523 to 536), but I think it should be.

- Figure 8: It’d be nice if the legend clearly stated which graph is which. I think Figure 8b is the distal recombination pattern, but I’m not 100% sure. It’d be great to have sample sizes on the graph too (n = 34 or 16 species, I think).

- It’s pretty clear in the M&M, but on line 419, it might be nice to mention that rintra is a single value per chromosome. On my first reading, I was thinking it was some sort of transformation of cM between genes...

- The analysis of gene distances is very thought provoking!

- Line 488 – What the heck is going on with fungi and animals! It’s certainly not necessary, but can the authors provide a quick description or explanation. They have piqued my curiousity.

Again, I do not consider any of my comments to be critical for publication, and I want to again congratulate the authors on a thorough and interesting study.

Reviewer #3: This manuscript by Brazier and Glémin uses a comparative approach to investigate variation in recombination landscapes in flowering plants. Their study used genetic map data from 665 chromosomes in 57 species of angiosperms. At the whole chromosomal level, they found a negative correlation between chromosome size and recombination rate (cM/Mb) with a strong species-specific effect. They also found that CO excess on chromosomes was more correlated with their relative size to other chromosomes in the genome rather than their absolute size, and that this effect was consistent across species. When investigating crossover landscapes, they found that landscapes were similar within species but strongly varied between species. CO rates were not uniform across chromosomes and were often more likely to occur at the distal ends of the chromosomes, with larger chromosomes tending to have a higher “periphery bias” of COs. However (as with most things in nature), this general pattern did have a number of exceptions. The authors then investigated the joint effect of telomeres and centromeres on CO distribution, finding the strongest support for a model that incorporated the effects of the telomere, centromere and one CO per chromosome. The authors found that recombination rate increased with gene density. Finally, the authors showed that genetic shuffling was positively correlated with linkage map length, and that there was a small negative effect of the periphery-bias ratio. These effects were slightly higher when modelling genetic shuffling in terms of gene distances. Whilst the investigations here are largely correlative rather than revealing mechanisms, this study provides a useful foundation for further investigation of broad drivers of recombination rate and landscape variation across a wide range of taxa.

This paper is the most comprehensive and well analysed that I have read on this topic, and generally it is well-written and well structured, particularly the introduction and discussion. I’m impressed by the sheer breadth of analyses. Nevertheless, there are parts of the methods & results that lack clarity, which in turn leads to issues with reproducibility. In particular, a lot of the statistical models are not well described – model structures should be made explicit in the methods and/or results, rather than providing a general text for statistical analyses at the end of the methods. I would emphasise that providing code and data (where possible) would improve these issues.

I had many comments and suggestions - those marked ** should be addressed by the authors in a revised version.

ABSTRACT/INTRODUCTION

Lines 32-33: The authors should be clearer what they mean by “relative size” here (i.e. relative to the rest of the genome) and why this result is interesting.

Lines 48-52: In the first sentence, I would add the term “crossing-over” or “crossover” here to set up the rest of the introduction. In the second sentence, I would briefly define landscape (i.e. variation in recombination rate along the chromosomes)

Lines 61 – 63: Indicate that you are defining assurance in this sentence.

Line 73: Can the authors briefly mention how recombination landscapes shape the distribution of TEs?

**Lines 98 – 100: I found this statement confusing, as I can’t understand how independence between linkage map length and genome size means that recombination rates will be higher in smaller genomes. I also can’t make the link between this statement and the Stapley paper – I think they found that linkage map lengths were smaller in smaller genomes, but also that chromosome number explained more variation (i.e. increased chromosome number lead to longer maps due to a higher minimum bound of recombination due to crossover assurance). Perhaps I am wrong, but regardless, it might be worth double-checking this statement and explaining it more clearly.

Line 100: on recombination rate, or landscape? Or both?

**Line 102: Based on your argument here, it is not clear how chromosome length links to biases of CO towards the peripheries – please clarify.

Line 117: Briefly define genetic shuffling and why it’s interesting – could even be mentioned earlier e.g. around lines 56 – 58.

RESULTS

**Lines 130 – 132: I don’t think this is described in the methods. Is there information on the number of progeny? I was curious about this but couldn’t find the information in the supplementary tables.

Line 143: This header could be interpreted that smaller chromosomes have more crossover events rather than more crossovers per unit length. Perhaps “Smaller chromosomes have higher recombination rates than larger ones”?

**Lines 153 – 169: Where are the methods for this LMER and what is the model structure? Is this what is being described in lines 831 – 850 of the methods? Throughout the paper, it needs to be clearer what models were run and what their fixed & random effect structures were in order to better interpret them.

Line 155: Does this mean that there is no/low phylogenetic signal of recombination rate?

Figure 1: This figure is busy. A suggestion for panel A: perhaps the dashed lines could be fit from axis to axis, to visually demarcate the 1 – 4CO expectations a bit better? For panel B, since this is the same data plotted twice, perhaps only the regression lines need to be visualised here rather than all of the points.

**Figure 2: I found this figure confusing. Some suggested edits:

Panel A could be wider to allow discerning of the slopes. I also struggled to understand what the isolines on the graph are showing even after reading several times. When using isolines, perhaps there is a need to define their values (as in Figure 1) – or perhaps they can be removed if making things too busy.

Panel B: I cannot interpret what this is showing – are there really lines in panel A that have intercepts of less than zero?

Panels B & C: I am very curious to see the error on these estimates.

Panels B & C: maybe these panels might be better suited in the supplementary?

Panel D: Again, very busy. Perhaps use of transparency of points or lines could make things clearer.

**Figure 4: Accessibility issue for colour-blindness - the red dots may not be visible on the green background. The visual scale for the chromosome size is a unclear, particularly as appears to be log – could there be line traces instead of colours here? Also – perhaps I have misunderstood – but if the chromosomes were split into ten bins, then why does the resolution of recombination rate estimation look to be much higher than 1/10th of the chromosome on the horizontal lines?

**Line 282 – 298: It seems that chromosome size was a strong correlate of recombination pattern, but I was curious if the authors tested other factors to rule out potential artefacts (e.g. differences in marker density) or to identify other biological correlates, such as ploidy? Was there a phylogenetic signal of this distal vs subdistal pattern?

Figure 6: There is a lot of text to wade through in the legend - it would help the reader to put annotations, sample sizes, key on the figures to allow for faster interpretation. For example, putting A: Distal pattern, N = XX, B: Subdistal pattern, N = XX on the panels make it easier to interpret. The dashed lines for the unclassified patterns are very distracting – why not include this as another panel, or put it in the supplementary material? Panel C is tiny and needs a key, or at least x-axis labels. I like the schematics of the crossover distributions but it’s so tiny – perhaps include this as its own figure as it explains the model really well.

Figure 7: A & B. There needs to be a higher contrast between the colours as it’s difficult to see the differences between blue and black. C. What do the colours represent here? Adjusting the point transparency and slight x jitter may improve the visualisation here.

Figure 8: Same as Figure 6 – annotating the panels would be helpful.

**Line 414 – I think it’s important for the authors to briefly define what genetic shuffling is and why it’s interesting to look at from an evolutionary perspective.

Line 419: On the same chromosome

Line 422: less efficient = resulted in less genomic shuffling?

Figure 9: see comments on Figure 7.

DISCUSSION

Lines 477 – 488: I’m a little puzzled by some of the statements here, so perhaps clarification is needed. I think it could be mentioned that crossover assurance will give a basal rate per chromosome of 50cM regardless of size, and then the authors can expand how the findings outlined here add to this established fact. Furthermore, I believe that in animals, larger chromosomes do have lower recombination rates within species… if I have misinterpreted this, perhaps the authors need to clarify their point better.

Line 519: clarify what “association” means here… does chromosome pairing begin at the telomeres?

Line 570: put “beam-film” in inverted commas and indicate that you are about to describe it.

Line 582 – 583: It depends on the number of gametes measured and how many were male and female, which is easily done in dioecious species. I think authors should specify here “in angiosperms” and iterate here why heterochiasmy is difficult to investigate for the less plant-literate reader.

METHODS

**Lines 700 – 704: Indicate that this was from cytogenetic data. How is this information orientated correctly to the linkage map/genome sequence?

**Lines 709 – 723: I think this paragraph requires a few improvements in reproducibility. Was this all done in the MareyMap package in the next paragraph? What was a ballpark criteria/example for anything that was outside the global trend?

**Lines 724 – 737: Related to the previous comment, when looking at the plotted Marey maps (Figure S1), are the methods/results affected in any way by the “jitter” of Mb vs cM distances? I imagine that if the markers were not in the correct order in the linkage map (if the local linkage order is ABC, but the real genomic order is ACB), then the cM length of the chromosome may be overestimated, meaning that recombination rates would be consistently inflated. For example, in Camellia sinensis, the maps seems to be messier and therefore may accumulate local overestimations in recombination rate that will lead to a longer cM map than the true one, compared to Arabidopsis where the orders appear to be highly conserved between the genome and the linkage map. The potential impact of this should be discussed.

**Lines 738 – 749: Please clarify here how the relative recombination rate is calculated – is this done for each segment? i.e. if the chromosome is 100cM and 80Mb, and the first segment is e.g. 10cM & 20Mb, then how would the value be calculated? The verbal argument is unclear.

**Lines 831 – 850: The way this is written, it isn’t connected to any specific models. It is important to describe what was modelled here and to be explicit about the model structures to ensure reproducibility.

**Have all data underlying the figures and results presented in the manuscript been provided?**

Reviewer #1: Yes

Reviewer #2: Yes

Reviewer #3: Yes

PLOS authors have the option to publish the peer review history of their article (what does this mean?). If published, this will include your full peer review and any attached files.

Reviewer #1: No

Reviewer #2: No

Reviewer #3: No

---

## [Decision Letter · Decision Letter 1]

20 Jul 2022

Dear Dr Brazier,

Thank you very much for submitting your Research Article entitled 'Diversity and determinants of recombination landscapes in flowering plants' to PLOS Genetics.

The manuscript was fully evaluated at the editorial level and by independent peer reviewers. The reviewers appreciated the attention to an important problem, but raised some substantial concerns about the current manuscript. Based on the reviews, we will not be able to accept this version of the manuscript, but we would be willing to review a much-revised version. We cannot, of course, promise publication at that time.

As you can see, two of your reviewers found your revised version acceptable for publication. However, reviewer 1 raises a number of concerns about the work and does not yet support acceptance, and they suggest multiple points for continued improvement of the manuscript.

Should you decide to revise the manuscript for further consideration here, your revisions should address the specific points made by reviewer 1. We will also require a detailed list of your responses to the review comments and a description of the changes you have made in the manuscript.

If you decide to revise the manuscript for further consideration at PLOS Genetics, please aim to resubmit within the next 60 days, unless it will take extra time to address the concerns of the reviewers, in which case we would appreciate an expected resubmission date by email to plosgenetics@plos.org.

[LINK]

We are sorry that we cannot be more positive about your manuscript at this stage. Please do not hesitate to contact us if you have any concerns or questions.

Yours sincerely,

Ian R. Henderson

Associate Editor

PLOS Genetics

Kirsten Bomblies

Section Editor: Evolution

PLOS Genetics

Reviewer's Responses to Questions

**Comments to the Authors:**

Reviewer #1: The revised manuscript still requires extensive revision of the English, which is frequently hard to understand (I attach an annotated pdf file as well as comments below). The Discussion also seems unduly long (7 pages) and could be shorter and clearer if unnecessary repetition of results were removed.

Another general comment is that the text frequently ignores the state of the art in the field. When a concept is already well-established, I feel that, if one cites a recent paper, the citation should make clear that this is a review of the topic. As written, without doing this, or citing the early literature, the manuscript misleadingly gives the impression that these things are new discoveries. My comments on Lines 56 and 127 below are in a category that I consider “not so minor”.

The chief question remains whether the present study represents an advance sufficient for acceptance by PLoS Genetics, as opposed to being better suited for a journal like G3 or Heredity. In other words, what new discovery does it provide?

It is correct that the size of the dataset is an improvement on previous analyses, and I consider it improvements in quality to describe recombination patterns, not just total map lengths, and to describe centromere locations (the value of such information was stressed in this recent paper, which should be cited: Yoshida, K., and J. Kitano, 2021 Tempo and mode in karyotype evolution revealed by a probabilistic model incorporating both chromosome number and morphology. PLoS Genetics 17: e1009502. doi: 10.1371/journal.pgen.1009502). Clearly, these points should be emphasised. My criticisms did not dispute the value of such improvements, or of re-analysing the data to make results comparable between species.

The authors’ responses make clear that several of the most interesting questions could not be studied, and it would be helpful to readers to mention these explicitly, perhaps in a brief “Future directions” section at the end. This would help readers understand how difficult it is to reach conclusions about these questions, and that the right kind of data are now obtainable, albeit with considerable effort.

The issue of family sizes appears not to be mentioned, but surely the ability to detect recombinants will depend on the family size, and, in small families, none may be detected in regions with infrequent crossing over even if crossovers can occasionally occur. At present, the genetic maps are taken as facts, not estimates. This problem extends beyond the issue of family sizes, and I believe that it also affects the analysis of the models. As crossovers happen in the 4-strand stage, the genetic map distances don’t directly tell us the number of crossovers per bivalent, and so it is difficult to infer anything clear about the number that is required for correct segregation. It might be nice to know whether the evidence supports a requirement for a crossover per chromosome, or whether this requirement is enforced for each chromosome arm. However, it is well-established that, despite some such requirement, crossover numbers show a distribution that includes zero events, yet correct segregation is possible.

However, my major criticism of the manuscript is still that the main new conclusion (line 434 onwards) does not seem justified. In my wording, this is that recombination between coding regions occurs more (“was more efficient”) than among regions randomly sampled from the genome, especially for longer chromosomes, which tend to have distal crossover localisation, and consequently the most heterogeneous recombination rates. The reasoning used to reach the conclusion that this is an important evolutionary observation seems to overlook an “elephant in the room” — the accumulation of repetitive sequences in low recombination regions of genomes. A direct outcome of this tendency is that regions with low recombination rates tend also have low gene densities. I don’t think that this has (in my edited wording) “implications for the evolution of crossover landscapes and whether the distribution of COs is optimal for the efficacy of genetic recombination”.

Following Haenel et al. (2018), the authors show that recombination rates correlate positively with gene density. This is unsurprising, as gene density goers down as repetitive content goes up, and the latter increases in genome regions with low crossover rates. This has been established in many species, including plants. Line 373 onwards says “the strength of the relationship greatly varied across species and did not correlate with usual predictors such as the chromosome length or the genome-wide recombination rate”. But surely the usual predictors of gene density are not chromosome length or the genome-wide recombination rate, but local recombination rates, with clear effects of proximity to centromeres. This is because these effects are pronounced only in genome regions with pretty low recombination rates, and cannot be detected from correlations using minor or local recombination rate differences.

By writing that recombination rates “consistently increased with the number of genes” in 100 kb windows, the text suggests in the readers mind that the causative factor is the number of genes (even though clearly a correlation does not imply such a causation, and of course I don’t think that the authors intended to create this misleading impression).

Marey maps most often appeared more homogeneous when the x axis measures distances in terms of cumulative number of genes along the chromosome instead of base pair distances. When genes are sparse, a small value of this “gene distance” corresponds to a large physical distance, so the inhomogeneity of the maps is diminished, as a direct arithmetical consequence (provided that the annotation of the genes is accurate). This has no special biological significance.

If I have misunderstood the authors’ reasoning, I am happy to be corrected. Certainly, the writing needs to make the meaning clearer if what I have understood them to be saying is not their meaning.

Not so minor

Lines 56 and 127: The text should not give the impression that an association between gene density and recombination is a new observation. Line 127 says “are recombination landscapes generally associated with gene density?” (meaning “are recombination RATEs generally associated with gene density?”), but such a relationship is already extremely well established, because it is known that repetitive sequences accumulate in low recombination regions of genomes. If this is to be mentioned in the Introduction, it needs to be changed to make clear that this well-known pattern is confirmed by these new analyses. The fact that recombination hotspots

have been found in gene regulatory sequences is a minor contributor to such a pattern.

Line 66: It is strange to cite de Massy, 2013 and later papers for the discovery that recombination rates are not homogeneous across the genome and vary among species, as this has been known for a very long time. Centromeric and pericentromeric regions with low rates were known for plants at least since work on maize in the 1930s, and work on tomatoes shortly afterwards, showing the heterochromatic regions physically surrounding centromeres are recombinationally inert.

Similarly, in line 294, it reads strangely that “we observed that the centromeres had an almost universal local suppressor effect”, as this is such a well-known phenomenon (since Beadle’s work in Drosophila in 1932), and is known to act in addition to the effect of heterochromatin. It would be better to write that thewell-known phenomenon of centromeric low crossover rates is confirmed. In this paragraph, a caveat should be expressed to the statement about “completely recombination-free in the centromere” (and also about the short genetic maps of some arms in line 331), as surely the ability to detect recombinants will depend on the family size, and none may be detected in small families, even if crossovers occasionally occur.

Line 192: Figure 1 shows that most of the 57 plant species analysed have estimated cM/Mb rates >1. This is a valuable result, but the text doesn’t mention it. I would have liked to see the value of 1 indicated by a horizontal line, as it is not shown on the y axis. From the figure, it appears that 4 species consistently have values < 0.5. Their names should be mentioned, as such low values are unusual, and these results should be examined to find out whether they are trustworthy, of if terminal markers may be lacking. Figure 2A seems to show that these same species may have very long chromosomes. This too seems worthy of explicit remark, as many readers will be interested to learn that plant chromosomes can be larger than 500 Mb (again, the species should be named, as perhaps these are species already known to have highly repetitive, and physically large, genomes). As presented, the reader does not even know whether these are related species (looking at Figure 4, I wonder if these are the 4 grass species at the bottom of that Figure?).

This comment exemplifies my previous concern that only relationships between quantities are studied, when in fact some of the values are of interest in themselves. Omitting to even mention them makes the findings hard to understand, as it is easy to “miss the wood for the trees”, or understand clearly what the complicated relationships might mean.

Line 222: the text is still not clear (“used the terms proximal and distal regions, respectively, to avoid confusion with the molecular composition and specific position defining telomeric and centromeric regions stricto sensu”). Do “proximal and distal regions” mean centromere-proximal versus distal positions (on both arms for metacentrics), respectively? (and I think that the correct phrase is “sensu strictu”). And does “middle” or “centre” refer to the middle of an arm in a metacentric, or to the centromeric region and its neighbourhood (whereas, for an acrocentric, it would refer to a region distant from the centromere). Clarification is still needed.

I also did not understand which landscapes were homogeneous along chromosomes — it would be helpful to name explicitly the species in Figure 3 that fall into the different categories, and mention the test for homogeneity that was used. The differences are not obvious to me.

Line 265 states that the confidence interval for the periphery bias is 2.06 - 2.32., indicating that recombination rates are highest in the tips of chromosomes., though the differences are not usually extreme. Confusingly, the text earlier in this section strongly emphasised that a bias towards the periphery was not ubiquitous across species. These things are not necessarily contradictory, but they give the impression of being so. In fact, if I have understood correctly, it appears from Figure 5 that Haenel et al. were correct in thinking that this is a common pattern. If so, this should be clearly stated in the text, as it is an important conclusion, and readers should not be left with the impression that they were wrong (even if there are a few exceptions to this pattern — in fact, the exceptions could be of interest).

Minor comments

A minor, but general, issue is the failure to distinguish between physical positions in the chromosome assemblies and genetic map positions. Although the text makes use of both these measures, it is sometimes unclear which one is meant , so that the meaning is not clear (e.g. lines 313 to 315).

It is simplistic to write that recombination “increases genetic diversity and the adaptive potential of a species”. Most readers will know that genetic recombination is interesting and worthy of study.

Line 51: what genomic characteristics? It would be better to say “genome size and ….” specifying the other characteristics.

Line 83: It is too sweeping to say that recombination hotspots are general. Some organisms don’t have them, so please write more cautiously. In addition, please make clear what definitions were used for the plant hotspots cited, as many authors use this term extremely loosely, to mean any region where they detect higher recombination than other regions, even if the regions are large, unlike the original definition of hotspots.

Line 87: it is too vague to just say that recombination affects genome structure, functioning and evolution. Please be explicit. For example, recombination events can affect genome structure by causing ectopic recombination between similar sequences, including repeats, in different locations. As written, readers will not understand what you have in mind.

Line 107: the meaning is unclear of “positively driven by chromosome length”

Line 119: the phrase “to our knowledge” is in the wrong place in the sentence.

Line 128: the question “What are the consequences of recombination heterogeneity on the extent of genetic shuffling?” is hard to understand, as “genetic shuffling” is usually just “baby language” for recombination (in my opinion, it should not be used in a scientific paper). Is this an attempt to mention that recombination can involve both crossing over and gene conversion? If so, it is certainly not clear enough. I would advise a different term for the sub-heading in line 405.

Line 169: Are the crossover numbers means? It is difficult to understand what is meant by “234 chromosomes had between one and two COs”, as a chromosome has a definite number. This, and the preceding sentence should be revised to make the meaning clear.

Line 173: The meaning of a “species random effect” is not clear enough.

Line 263: what does “ex on Fig 3A and 3E” mean?

Line 289: the meaning is unclear. The text reads “chromosomes from species classified as having a distal pattern were significantly larger than chromosomes with a sub-distal pattern”. Does this mean “species classified as having the distal pattern had significantly larger chromosomes than those of species assigned to the set with the sub-distal pattern”.

Line 313: The meaning is unclear for “equal distribution of crossovers on both sides of the

chromosome”,

Line 321: This reads “At least one CO in each chromosome arm (50 cM) is mandatory in M2 whereas only one CO is mandatory for the entire chromosome in M3”, which is difficult to understand. Does it mean “In M2, at least one CO in each chromosome arm (50 cM) is mandatory, whereas in M3 only a single CO is mandatory for the entire chromosome, even if it has two arms”?

Reviewer #2: I want to congratulate the authors again on an excellent paper. All my (minor) suggestions were met very aptly, and overall the paper is improved with the revisions.

Reviewer #3: Thank you for addressing our suggested revisions. In my opinion, this paper is much improved from the previous draft. It is so much easier to follow the different analyses and findings this time around. It made me really appreciate the discussion as an excellent synthesis of all the findings of this paper in a broader context. The figures look much better and do justice to all the work that is being presented.

A few small corrections:

• Throughout the Abstract/Summary/Intro, I would use “crossover” rather than “crossing-over” when referring to the process as a noun e.g. number of crossovers vs. during crossing-over (verb). (I realise I may not have been clear enough in my original review)

• Line 24: characterise

• Line 35: correspond globally

• Lines 63 – 64: might make more sense to say something like “New haplotypes are passed on to offspring by the reciprocal exchange of DNA between maternal and paternal chromosomes, known as crossovers (COs)” and then remove “(COs)” on line 74

• Line 68: specific pairing sites implies that recombination always takes place in the same spot

• Lines 84-85: naïve question for you to check – is it in regulatory regions rather that sequence? Promotor regions?

• Lines 172 – 174: can you get at how much of the variance species explains?

• Figure 6: Define pattern “E” in the legend

• For my comment about the local linkage order potentially inflating map length, I thought your response was good – could it be included in the methods in case another reader had the same concern?

**Have all data underlying the figures and results presented in the manuscript been provided?**

Reviewer #1: Yes

Reviewer #2: Yes

Reviewer #3: Yes

PLOS authors have the option to publish the peer review history of their article (what does this mean?). If published, this will include your full peer review and any attached files.

Reviewer #1: No

Reviewer #2: No

Reviewer #3: No

---

## [Editor Report · Decision Letter 2]

5 Aug 2022

Dear Dr Brazier,

We are pleased to inform you that your manuscript entitled "Diversity and determinants of recombination landscapes in flowering plants" has been editorially accepted for publication in PLOS Genetics. Congratulations!

Yours sincerely,

Ian R. Henderson

Academic Editor

PLOS Genetics

Kirsten Bomblies

Section Editor

PLOS Genetics

Comments from the reviewers (if applicable):

**Data Deposition**

http://datadryad.org/submit?journalID=pgenetics&manu=PGENETICS-D-22-00286R2

**Press Queries**

---

## [Editor Report · Acceptance letter]

24 Aug 2022

PGENETICS-D-22-00286R2 

Diversity and determinants of recombination landscapes in flowering plants 

Dear Dr Glemin, 

We are pleased to inform you that your manuscript entitled "Diversity and determinants of recombination landscapes in flowering plants" has been formally accepted for publication in PLOS Genetics! Your manuscript is now with our production department and you will be notified of the publication date in due course.

With kind regards,

Zsofi Zombor

PLOS Genetics

On behalf of:
